# FUNCTIONAL BAYESIAN TUCKER DECOMPOSITION FOR CONTINUOUS-INDEXED TENSOR DATA

**Shikai Fang, Xin Yu, Zheng Wang, Shibo Li, Robert M. Kirby, Shandian Zhe**[*]
University of Utah, Salt Lake City, UT 84112, USA
`{shikai, xiny, wzuht, shibo, kirby, zhe}@cs.utah.edu`

## ABSTRACT

Tucker decomposition is a powerful tensor model to handle multi-aspect data. It demonstrates the low-rank property by decomposing the grid-structured data as interactions between a core tensor and a set of object representations (factors). A fundamental assumption of such decomposition is that there are finite objects in each aspect or mode, corresponding to discrete indexes of data entries. However, real-world data is often not naturally posed in this setting. For example, geographic data is represented as continuous indexes of latitude and longitude coordinates, and cannot fit tensor models directly. To generalize Tucker decomposition to such scenarios, we propose Functional Bayesian Tucker Decomposition (FunBaT). We treat the continuous-indexed data as the interaction between the Tucker core and a group of latent functions. We use Gaussian processes (GP) as functional priors to model the latent functions. Then, we convert each GP into a state-space prior by constructing an equivalent stochastic differential equation (SDE) to reduce computational cost. An efficient inference algorithm is developed for scalable posterior approximation based on advanced message-passing techniques. The advantage of our method is shown in both synthetic data and several real-world applications. We release the code of FunBaT at `https://github.com/xuangu-fang/Functional-Bayesian-Tucker-Decomposition`

## 1 INTRODUCTION

Tensor decomposition is widely used to analyze and predict with multi-way or multi-aspect data in real-world applications. For example, medical service records can be summarized by a four-mode tensor (*patients*, *doctor*, *clinics*, *time*) and the entry values can be the visit count or drug usage. Weather data can be abstracted by tensors like (*latitude*, *longitude*, *time*) and the entry values can be the temperature. Tensor decomposition introduces a set of low-rank representations of objects in each mode, known as factors, and estimates these factors to reconstruct the observed entry values. Among numerous proposed tensor decomposition models, such as CANDECOMP/PARAFAC (CP) (Harshman, 1970) decomposition and tensor train (TT) (Oseledets, 2011), Tucker decomposition (Tucker, 1966) is widely-used for its fair capacity, flexibility and interpretability.

Despite its success, the standard tensor decomposition framework has a fundamental limitation. That is, it restricts the data format as structured grids, and demands each mode of the tensor must be discrete and finite-dimensional. That means, each mode includes a finite set of objects, indexed by integers, such as user 1, user 2, ..., and each entry value is located by the tuple of discrete indexes. However, in many real-world applications like climate and geographic modeling, the data is represented as continuous coordinates of modes, *e.g., latitude*, *longitude* and *time*. In these cases, each data entry is located by a tuple of continuous indexes, and tensor decomposition cannot be applied directly.

To enable tensor models on such data, one widely-used trick is to discretize the modes, for example, binning the timestamps into time steps (say, by weeks or months), or binning the latitude values into a set of ranges, and number the ranges by 1, 2, 3... While doing this is feasible, the fine-grained information is lost, and it is hard to decide an appropriate discretization strategy in many cases

---

[*]Corresponding author

(Pasricha et al., 2022). A more natural solution is to extend tensor models from grid-structure to continuous field. To the best of our knowledge, most existing methods in that direction are limited to tensor-train (Gorodetsky et al., 2015; Bigoni et al., 2016; Ballester-Ripoll et al., 2019; Chertkov et al., 2023) or the simpler CP format (Schmidt, 2009), and cannot be applied to Tucker decomposition, an important compact and flexible low-rank representation. What's more, most of the proposed methods are deterministic with polynomial approximations, cannot provide probabilistic inference, and are hard to well handle noisy, incomplete data.

To bridge the gap, we propose FunBaT: Functional Bayesian Tucker decomposition, which generalizes the standard Tucker decomposition to the functional field under the probabilistic framework. We decompose the continuous-indexed tensor data as the interaction between the Tucker core and a group of latent functions, which map the continuous indexes to mode-wise factors. We approximate such functions by Gaussian processes (GPs), and follow (Hartikainen and Särkkä, 2010) to convert them into state-space priors by constructing equivalent stochastic differential equations (SDE) to reduce computational cost. We then develop an efficient inference algorithm based on the conditional expectation propagation (CEP) framework (Wang and Zhe, 2019) and Bayesian filter to approximate the posterior distribution of the factors. We show this framework can also be used to extend CP decomposition to continuous-indexed data with a simplified setting. For evaluation, we conducted experiments on both synthetic data and real-world applications of climate modeling. The results show that FunBaT not only outperforms the state-of-the-art methods by large margins in terms of reconstruction error, but also identifies interpretable patterns that align with domain knowledge.

## 2 PRELIMINARY

### 2.1 TENSOR DECOMPOSITION AND FUNCTION FACTORIZATION

**Standard tensor decomposition** operates under the following setting: given a $K$-mode tensor $\mathcal{Y} \in \mathbb{R}^{d_1 \times \cdots \times d_K}$, where each mode $k$ contains $d_k$ objects, and each tensor entry is indexed by a $K$-size tuple $\mathbf{i} = (i_1, \ldots, i_K)$, denoted as $y_\mathbf{i}$. To decompose the tensor into a compact and low-rank form, we introduce $K$ groups of latent factors, denoted as $\mathcal{U} = \{\mathbf{U}^1, \ldots, \mathbf{U}^K\}$, where each $\mathbf{U}^k = [\mathbf{u}_1^k, \ldots, \mathbf{u}_{d_k}^k]^\top$. Each factor $\mathbf{u}_j^k$ represents the latent embedding of the $j$-th object in the $k$-th mode. The classic CP (Harshman, 1970) assumes each factor $\mathbf{u}_j^k$ is a $r$-dimensional vector, and each tensor entry is decomposed as the product-sum of the factors: $y_\mathbf{i} \approx \left(\mathbf{u}_{i_1}^1 \circ \ldots \circ \mathbf{u}_{i_K}^K\right)^\top \mathbf{1}$, where $\circ$ is the element-wise product. Tucker decomposition (Tucker, 1966) takes one more step towards a more expressive and flexible low-rank structure. It allows us to set different factor ranks $\{r_1, \ldots r_K\}$ for each mode, by introducing a core tensor $\mathcal{W} \in \mathbb{R}^{r_1 \times \cdots \times r_K}$, known as the Tucker core. It models the interaction as a weighted-sum over all possible cross-rank factor multiplication:

$$y_\mathbf{i} \approx \text{vec}(\mathcal{W})^\top \left(\mathbf{u}_{i_1}^1 \otimes \ldots \otimes \mathbf{u}_{i_K}^K\right), \tag{1}$$

where $\otimes$ is the Kronecker product and $\text{vec}(\cdot)$ is the vectorization operator. Tucker decomposition will degenerate to CP if we set all modes' ranks equal and $\mathcal{W}$ diagonal. tensor train (TT) decomposition (Oseledets, 2011) is another classical tensor method. TT sets the TT-rank $\{r_0, r_1, \ldots r_K\}$ firstly, where $r_0 = r_K = 1$. Then each factor $\mathbf{u}_j^k$ is defined as a $r_k \times r_{k+1}$ matrix. The tensor entry $y_\mathbf{i}$ is then decomposed as a series of the matrix product of the factors.

**Function factorization** refers to decomposing a complex multivariate function into a set of low-dimensional functions (Rai, 2014; Nouy, 2015). The most important and widely used tensor method for function factorization is TT (Gorodetsky et al., 2015; Bigoni et al., 2016; Ballester-Ripoll et al., 2019). It converts the function approximation into a tensor decomposition problem. A canonical form of applying TT to factorize a multivariate function $f(\mathbf{x})$ with $K$ variables $\mathbf{x} = (x_1, \ldots x_K) \in \mathbb{R}^K$ is:

$$f(\mathbf{x}) \approx G_1(x_1) \times G_2(x_2) \times \ldots \times G_K(x_K), \tag{2}$$

where each $G_k(\cdot)$ is a univariate matrix-valued function, takes the scaler $x_k$ as the input and output a matrix $G_k(x_k) \in \mathbb{R}^{r_{k-1} \times r_k}$. It is straightforward to see that $G_k(\cdot)$ is a continuous generalization of the factor $\mathbf{u}_j^k$ in standard TT decomposition.

## 2.2 Gaussian Process as state space model

**Gaussian process** (GP) (Rasmussen and Williams, 2006) is a powerful non-parametric Bayesian model to approximate functions. Formally, a GP is a prior distribution over function $f(\cdot)$, such that $f(\cdot)$ evaluated at any finite set of points $\mathbf{x} = \{x_1, \ldots, x_N\}$ follows a multivariate Gaussian distribution, denoted as $f \sim \mathcal{GP}(0, \kappa(x, x'))$, where $\kappa(x, x')$ is the covariance function, also known as the kernel, measuring the similarity between two points. One of the most widely used kernels is the Matérn kernel:

$$\kappa_{\text{Matérn}} = \sigma^2 \frac{\left(\frac{\sqrt{2\nu}}{\ell}||x - x'||^2\right)^\nu}{\Gamma(\nu)2^{\nu-1}} K_\nu \left(\frac{\sqrt{2\nu}}{\ell}||x - x'||^2\right) \tag{3}$$

where $\{\sigma^2, \ell, \nu, p\}$ are hyperparameters determining the variance, length-scale, smoothness, and periodicity of the function, $K_\nu$ is the modified Bessel function, and $\Gamma(\cdot)$ is the Gamma function.

Despite the great capacity of GP, it suffers from cubic scaling complexity $O(N^3)$ for inference. To overcome this limitation, recent work (Hartikainen and Särkkä, 2010) used spectral analysis to show an equivalence between GPs with stationary kernels and linear time-invariant stochastic differential equations (LTI-SDEs). Specifically, we can formulate a vector-valued latent state $\mathbf{z}(x)$ comprising $f(x)$ and its derivatives up to $m$-th order. The GP $f(x) \sim \mathcal{GP}(0, \kappa)$ is equivalent to the solution of an LTI-SDE defined as:

$$\mathbf{z}(x) = \left(f(x), \frac{\mathrm{d}f(x)}{\mathrm{d}x}, \ldots, \frac{\mathrm{d}f^m(x)}{\mathrm{d}x}\right)^\top, \quad \frac{\mathrm{d}\mathbf{z}(x)}{\mathrm{d}x} = \mathbf{F}\mathbf{z}(x) + \mathbf{L}w(x), \tag{4}$$

where $\mathbf{F}$ and $\mathbf{L}$ are time-invariant coefficients, and $w(x)$ is the white noise process with density $q_s$. At any finite collection of points $x_1 < \ldots < x_N$, the SDE in (4) can be further discretized as a Gaussian-Markov chain, also known as the state-space model, defined as:

$$p(\mathbf{z}(x)) = p(\mathbf{z}_1) \prod_{n=1}^{N-1} p(\mathbf{z}_{n+1}|\mathbf{z}_n) = \mathcal{N}(\mathbf{z}(x_1)|\mathbf{0}, \mathbf{P}_\infty) \prod_{n=1}^{N-1} \mathcal{N}(\mathbf{z}(x_{n+1})|\mathbf{A}_n\mathbf{z}(x_n), \mathbf{Q}_n) \tag{5}$$

where $\mathbf{A}_n = \exp(\mathbf{F}\Delta_n)$, $\mathbf{Q}_n = \int_{t_n}^{t_{n+1}} \mathbf{A}_n\mathbf{L}\mathbf{L}^\top\mathbf{A}_n^\top q_s \mathrm{d}t$, $\Delta_n = x_{n+1} - x_n$, and $\mathbf{P}_\infty$ is the steady-state covariance matrix which can be obtained by solving the Lyapunov equation (Lancaster and Rodman, 1995). All the above parameters in (4) and (5) are fully determined by the kernel $\kappa$ and the time interval $\Delta_n$. For the Matérn kernel (3) with smoothness $\nu$ being an integer plus a half, $\mathbf{F}$, $\mathbf{L}$ and $\mathbf{P}_\infty$ possess closed forms (Särkkä, 2013). Specifically, when $\nu = 1/2$, we have $\{m = 0, \mathbf{F} = -1/\ell, \mathbf{L} = 1, q_s = 2\sigma^2/\ell, \mathbf{P}_\infty = \sigma^2\}$; for $\nu = 3/2$, we have $m = 1, \mathbf{F} = (0, 1; -\lambda^2, -2\lambda), \mathbf{L} = (0; 1), q_s = 4\sigma^2\lambda^3, \mathbf{P}_\infty = (\sigma^2, 0; 0, \lambda^2\sigma^2)$, where $\lambda = \sqrt{3}/\ell$. With the state space prior, efficient $O(n)$ inference can be achieved by using classical Bayesian sequential inference techniques, like Kalman filtering and RTS smoothing (Hartikainen and Särkkä, 2010). Then the original function $f(x)$ is simply the first element of the inferred latent state $\mathbf{z}(x)$.

## 3 Model

### 3.1 Functional Tucker Decomposition with Gaussian Process

Despite the successes of tensor models, the standard tensor models are unsuitable for continuous-indexed data, such as the climate data with modes *latitude*, *longitude* and *time*. The continuity property encoded in the real-value indexes will be dropped while applying discretization. Additionally, it can be challenging to determine the optimal discretization strategy, and the trained model cannot handle new objects with never-seen indexes. The function factorization idea with tensor structure is therefore a more natural solution to this problem. However, the early work (Schmidt, 2009) with the simplest CP model, uses sampling-based inference, which is limited in capacity and not scalable to large data. The TT-based function factorization methods (Gorodetsky et al., 2015; Bigoni et al., 2016; Ballester-Ripoll et al., 2019; Chertkov et al., 2023) take deterministic approximation like Chebyshev polynomials or splines, which is hard to handle data noises or provide uncertainty quantification.

Compared to CP and TT, Tucker decomposition possesses compact and flexible low-rank representation. The Tucker core can capture more global and interpretable patterns of the data. Thus, we aim to

extend Tucker decomposition to continuous-indexed data to fully utilize its advantages as a Bayesian method, and propose FunBaT: Functional Bayesian Tucker decomposition.

Aligned with the setting of the function factorization (2), we factorize a $K$-variate function $f(\mathbf{i})$ in Tucker format (1) with preset latent rank:$\{r_1, \ldots r_K\}$:

$$f(\mathbf{i}) = f(i_1, \ldots i_K) \approx \text{vec}(\mathcal{W})^\top \left( \mathbf{U}^1(i_1) \otimes \ldots \otimes \mathbf{U}^K(i_K) \right) \tag{6}$$

where $\mathbf{i} = (i_1, \ldots i_K) \in \mathbb{R}^K$, and $\mathcal{W} \in \mathbb{R}^{r_1 \times \cdots \times r_K}$ is the Tucker core, which is the same as the standard Tucker decomposition. However, $\mathbf{U}^k(\cdot)$ is a $r_k$-size vector-valued function, mapping the continuous index $i_k$ of mode $k$ to a $r_k$-dimensional latent factor. We assign independent GP priors over each output dimension of $\mathbf{U}^k(\cdot)$, and model $\mathbf{U}^k(\cdot)$ as the stack of a group of univariate scalar functions. Specifically, we have:

$$\mathbf{U}^k(i_k) = [u_1^k(i_k), \ldots, u_{r_k}^k(i_k)]^T; \; u_j^k(i_k) \sim \mathcal{GP}\left(0, \kappa(i_k, i_k')\right), j = 1 \ldots r_k \tag{7}$$

where $\kappa(i_k, i_k') : \mathbb{R} \times \mathbb{R} \to \mathbb{R}$ is the covariance (kernel) function of the $k$-th mode. In this work, we use the popular Matérn kernel (3) for GPs over all modes.

## 3.2 State-space-prior and Joint Probabilities

Given $N$ observed entries of a $K$-mode continuous-indexed tensor $\mathcal{Y}$, denoted as $\mathcal{D} = \{(\mathbf{i}^n, y_n)\}_{n=1\ldots N}$, where $\mathbf{i}^n = (i_1^n \ldots i_K^n)$ is the tuple of continuous indexes, and $y_n$ is the entry values, we can assume all the observations are sampled from the target Tucker-format function $f(\mathbf{i})$ and Gaussian noise $\tau^{-1}$. Specifically, we define the likelihood $l_n$ as:

$$l_n \triangleq p(y_n | \{\mathbf{U}^k\}_{k=1}^K, \mathcal{W}, \tau) = \mathcal{N}\left(y_n \mid \text{vec}(\mathcal{W})^\top \left(\mathbf{U}^1(i_1^n) \otimes \ldots \otimes \mathbf{U}^K(i_K^n)\right), \tau^{-1}\right). \tag{8}$$

We then handle the functional priors over the $\{\mathbf{U}^k\}_{k=1}^K$. As introduced in Section 2.2, the dimension-wise GP with Matérn kernel over $u_j^k(i_k)$ is equivalent to an $M$-order LTI-SDE (4), and then we can discrete it as a state space model (5). Thus, for each mode's function $\mathbf{U}^k$, we concatenate the state space representation over all $r_k$ dimensions of $\mathbf{U}^k(i_k)$ to a joint state variable $\mathbf{Z}^k(i_k) = \text{concat}[\mathbf{z}_1^k(i_k), \ldots, \mathbf{z}_{r_k}^k(i_k)]$ for $\mathbf{U}^k$, where $\mathbf{z}_j^k(i_k)$ is the state variable of $u_j^k(i_k)$. We can build an ordered index set $\mathcal{I}_k = \{i_k^1 \ldots i_k^{N_k}\}$, which includes the $N_k$ unique indexes of mode-$k$'s among all observed entries in $\mathcal{D}$. Then, the state space prior over $\mathbf{U}^k$ on $\mathcal{I}_k$ is:

$$p(\mathbf{U}^k) = p(\mathbf{Z}^k) = p(\mathbf{Z}^k(i_k^1), \ldots, \mathbf{Z}^k(i_k^{N_k})) = p(\mathbf{Z}_1^k) \prod_{s=1}^{N_k-1} p(\mathbf{Z}_{s+1}^k | \mathbf{Z}_s^k), \tag{9}$$

$$\text{where } p(\mathbf{Z}_1^k) = \mathcal{N}(\mathbf{Z}^k(i_k^1) | \mathbf{0}, \tilde{\mathbf{P}}_\infty^k); \; p(\mathbf{Z}_{s+1}^k | \mathbf{Z}_s^k) = \mathcal{N}(\mathbf{Z}^k(i_k^{s+1}) | \tilde{\mathbf{A}}_s^k \mathbf{Z}^k(i_k^s), \tilde{\mathbf{Q}}_s^k). \tag{10}$$

The parameters in (10) are block-diagonal concatenates of the corresponding univariate case. Namely, $\tilde{\mathbf{P}}_\infty^k = \text{BlockDiag}(\mathbf{P}_\infty^k \ldots \mathbf{P}_\infty^k)$, $\tilde{\mathbf{A}}_s^k = \text{BlockDiag}(\mathbf{A}_s^k \ldots \mathbf{A}_s^k)$, and $\tilde{\mathbf{Q}}_s^k = \text{BlockDiag}(\mathbf{Q}_s^k \ldots \mathbf{Q}_s^k)$, where $\mathbf{P}_\infty^k$, $\mathbf{A}_s^k$ and $\mathbf{Q}_s^k$ are the corresponding parameters in (5). With $\mathbf{Z}^k(i_k)$, we can fetch the value of $\mathbf{U}^k$ by multiplying with a projection matrix $\mathbf{H} = \text{BlockDiag}([1, 0, \ldots], \ldots [1, 0, \ldots]): \mathbf{U}^k = \mathbf{H}\mathbf{Z}^k$.

With state space priors of the latent functions, we further assign a Gamma prior over the noise precision $\tau$ and a Gaussian prior over the Tucker core. For compact notation, we denote the set of all random variables as $\mathbf{\Theta} = \{\mathcal{W}, \tau, \{\mathbf{Z}^k\}_{k=1}^K\}$. Finally, the joint probability of the proposed model is:

$$p(\mathcal{D}, \mathbf{\Theta}) = p(\mathcal{D}, \{\mathbf{Z}^k\}_{k=1}^K, \mathcal{W}, \tau) = p(\tau)p(\mathcal{W}) \prod_{k=1}^K [p(\mathbf{Z}_1^k) \prod_{s=1}^{N_k-1} p(\mathbf{Z}_{s+1}^k | \mathbf{Z}_s^k)] \prod_{n=1}^N l_n, \tag{11}$$

where $p(\mathcal{W}) = \mathcal{N}(\text{vec}(\mathcal{W}) \mid \mathbf{0}, \mathbf{I})$, $p(\tau) = \text{Gam}(\tau | a_0, b_0)$, $a_0$ and $b_0$ are the hyperparameters of the Gamma prior, and $l_n$ is the data likelihood defined in (8).

## 4 Algorithm

The inference of the exact posterior $p(\mathbf{\Theta} | \mathcal{D})$ with (11) is intractable, as there are multiple latent functions $\{\mathbf{U}^k\}_{k=1}^K$ and the Tucker core interleave together in a complex manner in the likelihood

term $l_n$. To address this issue, we propose an efficient approximation algorithm, which first decouples the likelihood term by a factorized approximation, and then applies the sequential inference of Kalman filter and RTS smoother to infer the latent variables $\{\mathbf{Z}^k\}_{k=1}^K$ at each observed index. Finally, we employ the conditional moment matching technique to update the message factors in parallel. We will introduce the details in the following subsections.

## 4.1 Factorized Approximation with Gaussian and Gamma distribution

To estimate the intractable posterior $p(\mathbf{\Theta}|\mathcal{D})$ with a tractable $q(\mathbf{\Theta})$, we first apply the mean-field assumption and design the approximated posterior as a fully factorized format. Specifically, we approximate the posterior as:

$$p(\mathbf{\Theta}|\mathcal{D}) \approx q(\mathbf{\Theta}) = q(\tau)q(\mathcal{W})\prod_{k=1}^K q(\mathbf{Z}^k) \tag{12}$$

where $q(\tau) = \text{Gam}(\tau|a,b)$, $q(\mathcal{W}) = \mathcal{N}(\text{vec}(\mathcal{W}) \mid \boldsymbol{\mu}, \mathbf{S})$ are the approximated posterior of $\tau$ and $\mathcal{W}$, respectively. For $q(\mathbf{Z}^k)$, we further decompose it over the observed indexes set $\mathcal{I}_k$ as $q(\mathbf{Z}^k) = \prod_{s=1}^{N_k} q(\mathbf{Z}_s^k)$, where $q(\mathbf{Z}_s^k) = q(\mathbf{Z}^k(i_s^k)) = \mathcal{N}(\mathbf{Z}_s^k \mid \mathbf{m}_s^k, \mathbf{V}_s^k)$. Our goal is to estimate the variational parameters $\{a, b, \boldsymbol{\mu}, \mathbf{S}, \{\mathbf{m}_s^k, \mathbf{V}_s^k\}\}$, and make $q(\mathbf{\Theta})$ close to $p(\mathbf{\Theta}|\mathcal{D})$.

To do so, we use Expectation Propagation (EP) (Minka, 2001), to update $q(\mathbf{\Theta})$. However, the standard EP cannot work because the complex Tucker form of the likelihood term $l_n$ makes it intractable to compute the expectation of the likelihood term $l_n$ under $q(\mathbf{\Theta})$. Thus, we use the advanced moment-matching technique, Conditional Expectation Propagation (CEP) (Wang and Zhe, 2019) to address this issue. With CEP, we employ a factorized approximation to decouple the likelihood term $l_n$ into a group of message factors $\{f_n\}$:

$$\mathcal{N}\left(y_n \mid \text{vec}(\mathcal{W})^\top \left(\mathbf{U}^1(i_1^n) \otimes \ldots \otimes \mathbf{U}^K(i_K^n)\right), \tau^{-1}\right) \approx Z_n f_n(\tau) f_n(\mathcal{W}) \prod_{k=1}^K f_n(\mathbf{Z}^k(i_k^n)), \tag{13}$$

where $Z_n$ is the normalized constant, $f_n(\tau) = \text{Gam}(\tau|a_n, b_n)$, $f_n(\mathcal{W}) = \mathcal{N}(\text{vec}(\mathcal{W}) \mid \boldsymbol{\mu}_n, \mathbf{S}_n)$, $f_n(\mathbf{Z}^k(i_k^n)) = \mathcal{N}(\mathbf{Z}^k(i_k^n)|\mathbf{m}_n^k, \mathbf{V}_n^k)$ are the message factors obtained from the conditional moment-matching of $\mathbb{E}_q[l_n]$. Then we update the posterior $q(\tau)$ and $q(\mathcal{W})$ by simply merging the message factors from likelihood (13) and priors:

$$q(\tau) = p(\tau)\prod_{n=1}^N f_n(\tau) = \text{Gam}(\tau|a_0, b_0)\prod_{n=1}^N \text{Gam}(\tau|a_n, b_n), \tag{14}$$

$$q(\mathcal{W}) = p(\mathcal{W})\prod_{n=1}^N f_n(\mathcal{W}) = \mathcal{N}(\text{vec}(\mathcal{W}) \mid \mathbf{0}, \mathbf{I})\prod_{n=1}^N \mathcal{N}(\text{vec}(\mathcal{W}) \mid \boldsymbol{\mu}_n, \mathbf{S}_n). \tag{15}$$

The message approximation in (13) and message merging (14)(15) are based on the conditional moment-matching technique and the property of exponential family presented in (Wang and Zhe, 2019). All the involved computation is closed-form and can be conducted in parallel. We provide the details along with the introduction of EP and CEP in the appendix.

## 4.2 Sequential State Inference with Bayesian Filter and Smoother

Handling $q(\mathbf{Z}^k)$ is a bit more challenging. It is because the prior of $\mathbf{Z}^k$ has a chain structure (9)(10), and we need to handle complicated integration over the whole chain to obtain marginal posterior $q(\mathbf{Z}_s^k)$ in the standard inference. However, with the classical Bayesian filter and smoother method, we can infer the posterior efficiently. Specifically, the state space structure over $\mathbf{Z}_s^k$ is:

$$q(\mathbf{Z}_s^k) = q(\mathbf{Z}_{s-1}^k)p(\mathbf{Z}_s^k|\mathbf{Z}_{s-1}^k)\prod_{n \in \mathcal{D}_s^k} f_n(\mathbf{Z}_s^k), \tag{16}$$

where $p(\mathbf{Z}_s^k|\mathbf{Z}_{s-1}^k)$ is the transitions of the state given in (10), and $\mathcal{D}_s^k$ is the subset of $\mathcal{D}$, including the observation entries whose $k$-mode index is $i_k^s$, namely, $\mathcal{D}_s^k = \{n : i_k^n = i_k^s \mid \mathbf{i}_n \in \mathcal{D}\}$. If we treat $\prod_{n \in \mathcal{D}_s^k} f_n(\mathbf{Z}_s^k)$, a group of Gaussian message factors, as the observation of the state space model, (16) is the standard Kalman Filter(KF) (Kalman, 1960). Thus, we can run the KF algorithm and compute $q(\mathbf{Z}_s^k)$ from $s = 1$ to $N_k$ sequentially. After the forward pass over the states, we can run RTS smoothing (Rauch et al., 1965) as a backward pass to compute the global posterior of $q(\mathbf{Z}_s^k)$. This sequential inference is widely used to infer state space GP models (Hartikainen and Särkkä, 2010; Särkkä, 2013).

---

**Algorithm 1** FunBaT

---

**Input:** Observations $\mathcal{D}$ of a $K$-mode continuous-indexed tensor , kernel hyperparameters, sorted unique indexes set $\{\mathcal{I}_k\}$ of each mode.
Initialize approx. posterior $q(\tau)$, $q(\mathcal{W})$, $\{q(\mathbf{Z}^k)\}$ and message factors for each likelihood.
**repeat**
    **for** $k = 1$ **to** $K$ **do**
        Approximate messages factors $\{f_n(\mathbf{Z}^k(i_k^n)), f_n(\tau), f_n(\mathcal{W})\}_{n=1}^N$ in parallel with CEP (13)
        Update the approximated posterior $q(\tau)$, $q(\mathcal{W})$ by merging the message (14)(15).
        Update $q(\mathbf{Z}^k)$ sequentially by KF and RTS smoother based on (16)
    **end for**
**until** Convergence
**Return:** $q(\tau), q(\mathcal{W}), \{q(\mathbf{Z}^k)\}_{k=1}^K$

---

The whole inference algorithm is organized as follows: with the observation $\mathcal{D}$, we initialize the approximated posteriors and message factors for each likelihood. Then for each mode, we firstly approximate the message factors by CEP in parallel, and then merge them to update $q(\tau), q(\mathcal{W})$. We run the KF and RTS smoother to infer $q(\mathbf{Z}^k)$ at each observed index by treating the message factors as observations. We repeat the inference until convergence. We summarize the algorithm in Table 1.

**Algorithm Complexity.** The overall time complexity is $\mathcal{O}(NKR)$, where $K$ is the number of modes, $N$ is the number of observations and $R$ is pre-set mode rank. The space complexity is $\mathcal{O}(NK(R + R^2))$, as we need to store all the message factors. The linear time and space complexity w.r.t both data size and tensor mode show the promising scalability of our algorithm.

**Probabilistic Interpolation at Arbitrary Index.** Although the state-space functional prior of $\mathbf{U}^k$ or $\mathbf{Z}^k$ is defined over finite observed index set $\mathcal{I}_k$, we highlight that we can handle the probabilistic interpolation of $\mathbf{Z}^k$ at arbitrary index $i_k^* \notin \mathcal{I}_k$ after model inference. Specifically, with $i_k^{s-1} < i_k^* < i_k^s$ we can infer the $q(\mathbf{Z}^k(i_k^*))$ by integrating the messages from the transitions of its neighbors and will obtain a closed-form solution:

$$q(\mathbf{Z}^k(i_k^*)) = \int q(\mathbf{Z}_{s-1}^k)p(\mathbf{Z}^k(i_k^*)|\mathbf{Z}_{s-1}^k)\mathrm{d}\mathbf{Z}_{s-1}^k \int q(\mathbf{Z}_s^k)p\left(\mathbf{Z}_s^k|\mathbf{Z}^k(i_k^*)\right)\mathrm{d}\mathbf{Z}_s^k = \mathcal{N}(\mathbf{m}^*, \mathbf{V}^*) \quad (17)$$

We leave the detailed derivation in the appendix. This enables us to build a continuous trajectory for each mode, and predict the tensor value at any indexes, for which standard discrete-mode tensor decomposition cannot do.

**Lightweight Alternative: FunBaT-CP.** As the CP decomposition is a special and simplified case of Tucker decomposition, it is straightforward to build a functional CP decomposition with the proposed model and algorithm. Specifically, we only need to set the Tucker core $\mathcal{W}$ as all-zero constant tensor except diagonal elements as one, skip the inference step of $\mathcal{W}$, and perform the remaining steps. We then achieve FunBaT-CP, a simple and efficient functional Bayesian CP model. In some experiments, we found FunBaT-CP is more robust than FunBaT, as the dense Tucker core takes more parameters and is easier to get overfitting. We will show it in the experiment section.

## 5 RELATED WORK

The early work to apply tensor format to function factorization is (Schmidt, 2009). It takes the wrapped-GP to factorize the function with CP format and infers with Monte Carlo sampling, which is not scalable to large data. Applying tensor-train(TT) to approximate the multivariate function with low-rank structure is a popular topic (Gorodetsky et al., 2015; Bigoni et al., 2016; Ballester-Ripoll et al., 2019; Chertkov et al., 2022; 2023), with typical applications in simultaneous localization and mapping (SLAM) (Aulinas et al., 2008). These methods mainly use Chebyshev polynomials and splines as function basis, and rely on complex optimization methods like alternating updates, cross-interpolation (Gorodetsky et al., 2015; Bigoni et al., 2016) and ANOVA (analysis of variance) representation (Ballester-Ripoll et al., 2019; Chertkov et al., 2023). Despite the compact form, they are purely deterministic, sensitive to initial values and data noises, and cannot provide probabilistic inference. Fang et al. (2022; 2024) used similar techniques like state-space GP and CEP to our work, but they are designed to capture temporal information in tensor data. More discussions on the existing literature can be found in the appendix.

## 6 EXPERIMENT

### 6.1 SYNTHETIC DATA

We first evaluated FunBaT on a synthetic task by simulating a rank-1 two-mode tensor with each mode designed as a continuous function:

$$\mathbf{U}^1(i_1) = \exp(-2i_1) \cdot \sin(\frac{3}{2}\pi i_1); \mathbf{U}^2(i_2) = \sin^2(2\pi i_2) \cdot \cos(2\pi i_2). \tag{18}$$

The ground truth of the continuous-mode tensor, represented as a surface in Figure 1a, was obtained by: $y_{\mathbf{i}} = \mathbf{U}^1(i_1)\mathbf{U}^1(i_2)$. We randomly sampled 650 indexes entries from $[0, 1] \times [0, 1]$ and added Gaussian noise $\epsilon \sim \mathcal{N}(0, 0.02)$ as the observations. We used PyTorch to implement FunBaT, which used the Matérn kernel with $\nu = 3/2$, and set $l = 0.1$ and $\sigma^2 = 1$. We trained the model with $R = 1$ and compared the learned continuous-mode functions with their ground truth. Our learned trajectories, shown in Figure 1c and Figure 1d clearly revealed the real mode functions. The shaded area is the estimated standard deviation. Besides, we used the learned factors and the interpolation (17) to reconstruct the whole tensor surface, shown in Figure 1b. The numerical results of the reconstruction with different observed ratios can be found in the appendix.

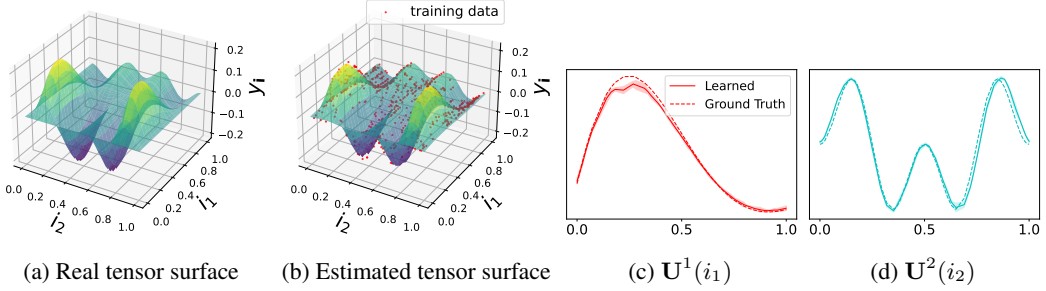

(a) Real tensor surface    (b) Estimated tensor surface    (c) $\mathbf{U}^1(i_1)$    (d) $\mathbf{U}^2(i_2)$

Figure 1: Results of Synthetic Data

### 6.2 REAL-WORLD APPLICATIONS

**Datasets** We evaluated FunBaT on four real-world datasets: *BeijingAir-PM2.5*, *BeijingAir-PM10*, *BeijingAir-SO2* and *US-TEMP*. The first three are extracted from BeijingAir[1], which contain hourly measurements of several air pollutants with weather features in Beijing from 2014 to 2017. We selected three continuous-indexed modes: *(atmospheric-pressure, temperature, time)* and processed it into three tensors by using different pollutants (PM2.5, PM10, SO2). Each dataset contains $17K$ observations across unique indexes of 428 atmospheric pressure measurements, 501 temperature measurements, and 1461 timestamps. We obtain *US-TEMP* from the ClimateChange[2]. The dataset contains temperatures of cities worldwide and geospatial features. We selected temperature data from 248 cities in the United States from 1750 to 2016 with three continuous-indexed modes: *(latitude, longitude, time)*. The tensor contains $56K$ observations across 15 unique latitudes, 95 longitudes, and 267 timestamps.

**Baselines and Settings** We set three groups of methods as baselines. The first group includes the state-of-art standard Tucker models, including *P-Tucker* (Oh et al., 2018): a scalable Tucker algorithm that performs parallel row-wise updates, *Tucker-ALS*: Efficient Tucker decomposition algorithm using alternating least squares(ALS) update (Bader and Kolda, 2008), and *Tucker-SVI* (Hoffman et al., 2013): Bayesian version of Tucker updating with stochastic variational inference(SVI). The second group includes functional tensor-train(FTT) based methods with different functional basis and optimization strategy: *FTT-ALS* (Bigoni et al., 2016), *FTT-ANOVA* (Ballester-Ripoll et al., 2019), and *FTT-cross* (Gorodetsky et al., 2015). As we can view the prediction task as a regression problem and use the continuous indexe as features, we add *RBF-SVM*: Support Vector Machine (Hearst et al., 1998) with RBF kernels, and *BLR*: Bayesian Linear Regression (Minka, 2000) as competing methods as the third group.

---

[1]https://archive.ics.uci.edu/ml/datasets/Beijing+Multi-Site+Air-Quality+Data

[2]https://berkeleyearth.org/data/

| | RMSE | | | MAE | | |
|---|---|---|---|---|---|---|
| Datasets | *PM2.5* | *PM10* | *SO2* | *PM2.5* | *PM10* | *SO2* |
| **Resolution: $50 \times 50 \times 150$** | | | | | | |
| P-Tucker | $0.805 \pm 0.017$ | $0.787 \pm 0.006$ | $0.686 \pm 0.02$ | $0.586 \pm 0.003$ | $0.595 \pm 0.005$ | $0.436 \pm 0.011$ |
| Tucker-ALS | $1.032 \pm 0.049$ | $1.005 \pm 0.029$ | $0.969 \pm 0.027$ | $0.729 \pm 0.016$ | $0.741 \pm 0.007$ | $0.654 \pm 0.034$ |
| Tucker-SVI | $0.792 \pm 0.01$ | $0.8 \pm 0.026$ | $0.701 \pm 0.08$ | $0.593 \pm 0.01$ | $0.605 \pm 0.019$ | $0.423 \pm 0.031$ |
| **Resolution: $100 \times 100 \times 300$** | | | | | | |
| P-Tucker | $0.8 \pm 0.101$ | $0.73 \pm 0.021$ | $0.644 \pm 0.023$ | $0.522 \pm 0.011$ | $0.529 \pm 0.013$ | $0.402 \pm 0.008$ |
| Tucker-ALS | $1.009 \pm 0.027$ | $1.009 \pm 0.026$ | $0.965 \pm 0.023$ | $0.738 \pm 0.01$ | $0.754 \pm 0.007$ | $0.68 \pm 0.011$ |
| Tucker-SVI | $0.706 \pm 0.011$ | $0.783 \pm 0.067$ | $0.69 \pm 0.086$ | $0.509 \pm 0.008$ | $0.556 \pm 0.031$ | $0.423 \pm 0.031$ |
| **Resolution: $300 \times 300 \times 1000$** | | | | | | |
| P-Tucker | $0.914 \pm 0.126$ | $1.155 \pm 0.001$ | $0.859 \pm 0.096$ | $0.401 \pm 0.023$ | $0.453 \pm 0.002$ | $0.366 \pm 0.015$ |
| Tucker-ALS | $1.025 \pm 0.044$ | $1.023 \pm 0.038$ | $1.003 \pm 0.019$ | $0.742 \pm 0.011$ | $0.757 \pm 0.011$ | $0.698 \pm 0.007$ |
| Tucker-SVI | $1.735 \pm 0.25$ | $1.448 \pm 0.176$ | $1.376 \pm 0.107$ | $0.76 \pm 0.033$ | $0.747 \pm 0.028$ | $0.718 \pm 0.023$ |
| **Resolution: $428 \times 501 \times 1461$ (original)** | | | | | | |
| P-Tucker | $1.256 \pm 0.084$ | $1.397 \pm 0.001$ | $0.963 \pm 0.169$ | $0.451 \pm 0.017$ | $0.493 \pm 0.001$ | $0.377 \pm 0.019$ |
| Tucker-ALS | $1.018 \pm 0.034$ | $1.012 \pm 0.021$ | $0.997 \pm 0.024$ | $0.738 \pm 0.005$ | $0.756 \pm 0.007$ | $0.698 \pm 0.011$ |
| Tucker-SVI | $1.891 \pm 0.231$ | $1.527 \pm 0.107$ | $1.613 \pm 0.091$ | $0.834 \pm 0.032$ | $0.787 \pm 0.018$ | $0.756 \pm 0.014$ |
| **Methods using continuous indexes** | | | | | | |
| FTT-ALS | $1.020 \pm 0.013$ | $1.001 \pm 0.013$ | $1.001 \pm 0.026$ | $0.744 \pm 0.007$ | $0.755 \pm 0.007$ | $0.696 \pm 0.011$ |
| FTT-ANOVA | $2.150 \pm 0.033$ | $2.007 \pm 0.015$ | $1.987 \pm 0.036$ | $1.788 \pm 0.031$ | $1.623 \pm 0.014$ | $1.499 \pm 0.018$ |
| FTT-cross | $0.942 \pm 0.025$ | $0.933 \pm 0.012$ | $0.844 \pm 0.026$ | $0.566 \pm 0.018$ | $0.561 \pm 0.011$ | $0.467 \pm 0.033$ |
| RBF-SVM | $0.995 \pm 0.015$ | $0.955 \pm 0.02$ | $0.794 \pm 0.026$ | $0.668 \pm 0.008$ | $0.674 \pm 0.014$ | $0.486 \pm 0.026$ |
| BLR | $0.998 \pm 0.013$ | $0.977 \pm 0.014$ | $0.837 \pm 0.021$ | $0.736 \pm 0.007$ | $0.739 \pm 0.008$ | $0.573 \pm 0.009$ |
| FunBaT-CP | $0.296 \pm 0.018$ | $0.343 \pm 0.028$ | $\mathbf{0.386 \pm 0.009}$ | $\mathbf{0.18 \pm 0.002}$ | $0.233 \pm 0.013$ | $0.242 \pm 0.003$ |
| FunBaT | $\mathbf{0.288 \pm 0.008}$ | $\mathbf{0.328 \pm 0.004}$ | $\mathbf{0.386 \pm 0.01}$ | $0.183 \pm 0.006$ | $\mathbf{0.226 \pm 0.002}$ | $\mathbf{0.241 \pm 0.004}$ |

Table 1: Prediction error over *BeijingAir-PM2.5*, *BeijingAir-PM10*, and *BeijingAir-SO2* with $R = 2$, which were averaged over five runs. The results for $R = 3, 5, 7$ are in the supplementary.

| | RMSE | | | MAE | | |
|---|---|---|---|---|---|---|
| Mode-Rank | R=3 | R=5 | R=7 | R=3 | R=5 | R=7 |
| P-Tucker | $1.306 \pm 0.02$ | $1.223 \pm 0.022$ | $1.172 \pm 0.042$ | $0.782 \pm 0.011$ | $0.675 \pm 0.014$ | $0.611 \pm 0.007$ |
| Tucker-ALS | $> 10$ | $> 10$ | $> 10$ | $> 10$ | $> 10$ | $> 10$ |
| Tucker-SVI | $1.438 \pm 0.025$ | $1.442 \pm 0.021$ | $1.39 \pm 0.09$ | $0.907 \pm 0.005$ | $0.908 \pm 0.005$ | $0.875 \pm 0.072$ |
| FTT-ALS | $1.613 \pm 0.0478$ | $1.610 \pm 0.052$ | $1.609 \pm 0.055$ | $0.967 \pm 0.009$ | $0.953 \pm 0.007$ | $0.942 \pm 0.010$ |
| FTT-ANOVA | $5.486 \pm 0.031$ | $4.619 \pm 0.054$ | $3.856 \pm 0.059$ | $4.768 \pm 0.026$ | $4.026 \pm 0.100$ | $3.123 \pm 0.0464$ |
| FTT-cross | $1.415 \pm 0.0287$ | $1.312 \pm 0.023$ | $1.285 \pm 0.052$ | $0.886 \pm 0.011$ | $0.822 \pm 0.006$ | $0.773 \pm 0.014$ |
| RBF-SVM | $2.374 \pm 0.047$ | $2.374 \pm 0.047$ | $2.374 \pm 0.047$ | $1.44 \pm 0.015$ | $1.44 \pm 0.015$ | $1.44 \pm 0.015$ |
| BLR | $2.959 \pm 0.041$ | $2.959 \pm 0.041$ | $2.959 \pm 0.041$ | $2.029 \pm 0.011$ | $2.029 \pm 0.011$ | $2.029 \pm 0.011$ |
| FunBaT-CP | $\mathbf{0.805 \pm 0.06}$ | $\mathbf{0.548 \pm 0.03}$ | $\mathbf{0.551 \pm 0.048}$ | $\mathbf{0.448 \pm 0.06}$ | $\mathbf{0.314 \pm 0.005}$ | $\mathbf{0.252 \pm 0.008}$ |
| FunBaT | $1.255 \pm 0.108$ | $1.182 \pm 0.117$ | $1.116 \pm 0.142$ | $0.736 \pm 0.069$ | $0.647 \pm 0.05$ | $0.572 \pm 0.089$ |

Table 2: Prediction error of *US-TEMP*, which were averaged over five runs.

We used the official open-source implementations of most baselines. We use the *TENEVA* library (Chertkov et al., 2022; 2023) to test the FTT-based methods. We re-scaled all continuous-mode indexes to $[0, 1]$ to ensure numerical robustness. For FunBaT, we varied Matérn kernels $\nu = \{1/2, 3/2\}$ along the kernel parameters for optimal performance for different datasets. We examined all the methods with rank $R \in \{2, 3, 5, 7\}$. We set all modes' ranks to $R$. Following (Tillinghast et al., 2020), we randomly sampled $80\%$ observed entry values for training and then tested on the remaining. We repeated the experiments five times, and examined the average root mean-square-error (RMSE), average mean-absolute-error (MAE), and their standard deviations.

To demonstrate the advantages of using continuous indexes rather than index discretization, we set four different discretization granularities by binding the raw continuous indexes into several discrete-indexed bins on *BeijingAir-PM2.5*, *BeijingAir-PM10*, and *BeijingAir-SO2*. We then derived

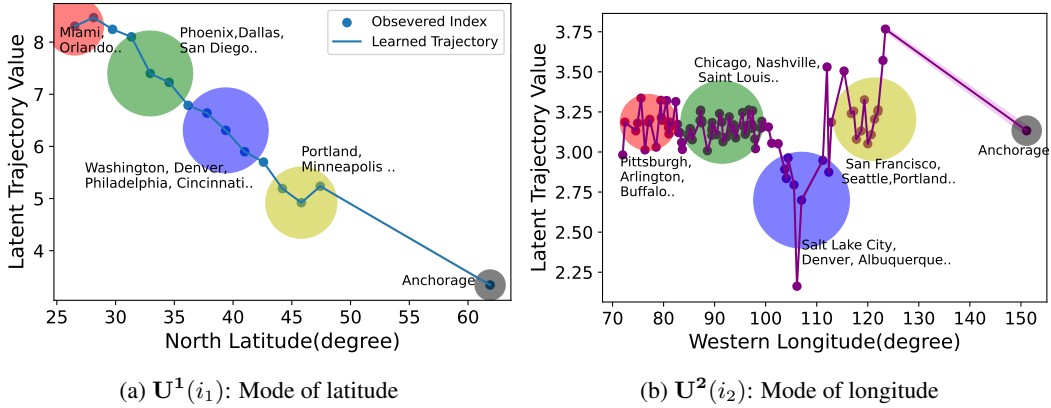

(a) $\mathbf{U}^1(i_1)$: Mode of latitude        (b) $\mathbf{U}^2(i_2)$: Mode of longitude

Figure 2: Learned mode functions of *US-TEMP*.

the tensors with four different resolutions: $428 \times 501 \times 1461$ (original resolution), $300 \times 300 \times 1000$, $100 \times 100 \times 300$, and $50 \times 50 \times 150$. We tested the performance of the first group (standard tensor decomposition) baselines on different resolutions. We tested FunBaT, FunBaT-CP, and other baselines with continuous indexes at the original resolution.

**Prediction Results.** Due to space limit, we only present the results with $R = 2$ for *BeijingAir-PM2.5*, *BeijingAir-PM10*, and *BeijingAir-SO2* datasets in Table 1, while the other results are in the appendix. The prediction results for *US-TEMP* are listed in Table 2. Our approach FunBaT and FunBaT-CP outperform the competing methods by a significant margin in all cases. We found the performance of standard Tucker methods varies a lot with different discrete resolutions, showing the hardness to decide the optimal discretization in real-world applications. We also found that the performance of FunBaT-CP is much better than FunBaT on *US-TEMP*. This might be because the dense Tucker core of FunBaT result in overfitting and is worse for the sparse *US-TEMP* dataset.

**Investigation of Learned Functions** We explored whether the learned mode functions reveal interpretable patterns that are consistent with domain knowledge. We run FunBaT on *US-TEMP* dataset with $R = 1$, and plotted the learned latent functions of the three modes *(latitude, longitude, time)*, shown in Figure 2 and 3. As the first two modes correspond to latitude and longitude, respectively, representing real geo-locations, we marked out the city groups (the shaded circles) and some city names in Figure 2a and 2b. The $\mathbf{U}^1$ of the lantitude mode in Figure 2a shows a clear decreasing trend from southern cities like Miami (in Florida) to northern cities like Anchorage (in Alaska), which is consistent with the fact that temperatures

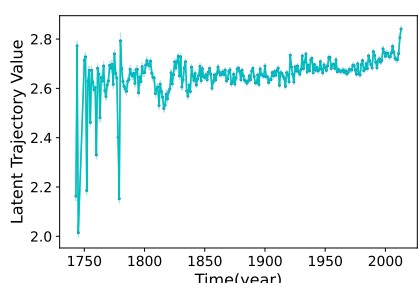

Figure 3: $\mathbf{U}^3(i_3)$: Mode of time

are generally higher in the south and lower in the north. The learned longitude-mode $\mathbf{U}^2$ in Figure 2b exhibits a steady trend from east to west, but with a region (the blue circle) with lower values near the Western longitude $110°$. That is the *Rocky Mountain Area* including cities like Denver and Salt Lake City with higher altitudes, and results in lower temperatures. The time-mode $\mathbf{U}^3$ in Figure 3 provides meaningful insights of the climate change in history. It reveals a gradual increase over the past 260 years, with a marked acceleration after 1950. This pattern aligns with the observed trend of rising global warming following the industrialization of the mid-20th century. The function around 1750-1770, characterized by lower and oscillating values, corresponds to the *Little Ice Age*, a well-documented period of global cooling in history.

## 7 CONCLUSION

We proposed FunBaT, a Bayesian method to generalize Tucker decomposition to the functional field to model continuous-indexed tensor data. We adopt the state-space GPs as functional prior and develop an efficient inference algorithm based on CEP. The results on both synthetic and real-world tasks demonstrate the effectiveness of our method.

## ACKNOWLEDGMENTS

This work has been supported by MURI AFOSR grant FA9550-20-1-0358, NSF CAREER Award IIS-2046295, and NSF OAC-2311685.

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

## A  CONDITIONAL EXPECTATION PROPROGATION

### A.1  BREIF INTRODUCTION OF EXPECTATION PROPROGATION(EP)

As a general framework of Bayesian learning, expectation proprogation (EP) (Minka, 2001) is a variational inference method for computing the posterior distribution of a latent variable $\boldsymbol{\theta}$ given the observed data $\mathcal{D}$. The exact posterior $p(\boldsymbol{\theta} \mid \mathcal{D})$ can be formed as following form:

$$p(\boldsymbol{\theta} \mid \mathcal{D}) = \frac{1}{Z} \prod_n p_n(\mathcal{D}_n \mid \boldsymbol{\theta}), \tag{19}$$

where $p_n(\mathcal{D}_n \mid \boldsymbol{\theta})$ is the likelihood of $n$-th data given $\boldsymbol{\theta}$, which may also link to the priors. $Z$ is the normalizer to ensure (19) is a valid distribution. The exact computation of (19) is often infeasible due to the complex form of $p_n(\mathcal{D}_n \mid \boldsymbol{\theta})$. Therefore, EP naturally adopts a factorized distribution approximation of real posterior:

$$p(\boldsymbol{\theta} \mid \mathcal{D}) \approx q(\boldsymbol{\theta} \mid \mathcal{D}) \propto \prod_n f_n(\boldsymbol{\theta}), \tag{20}$$

where $f_n(\boldsymbol{\theta})$ is an approximated factor for the actual data likelihood factor $p_n(\mathcal{D}_n \mid \boldsymbol{\theta})$. If we can assign a proper form of $f_n(\boldsymbol{\theta})$, and make $f_n(\boldsymbol{\theta}) \approx p_n(\mathcal{D}_n \mid \boldsymbol{\theta})$ as close as possible, then we can obtain a good approximation of the posterior distribution. The EP algorithm sets the basic form of factors using the exponential-family distribution, which includes many distributions such as Gaussian, Gamma, and Beta, and has the canonical formula $f_n(\boldsymbol{\theta}) \sim \exp(\boldsymbol{\lambda}_n^\top \phi(\boldsymbol{\theta}))$, where $\boldsymbol{\lambda}_n^\top$ and $\phi(\boldsymbol{\theta})$ are regarded as natural parameter and sufficient statistics of $f_n(\boldsymbol{\theta})$, respectively. EP iteratively updates the parameters of each $f_n(\boldsymbol{\theta})$ until convergence. The procedure to update each $f_n(\boldsymbol{\theta})$ is as follows:

(1) Building a calibrating distribution (a.k.s context factors) by removing current approx. factor :

$$q^{\backslash n}(\boldsymbol{\theta}) \propto q(\boldsymbol{\theta})/f_n(\boldsymbol{\theta}) \tag{21}$$

(2) Building a tilted distribution by multiplying back the real data likelihood with context factors:

$$\hat{p}_n(\boldsymbol{\theta}) \sim p_n(\mathcal{D}_n \mid \boldsymbol{\theta})q^{\backslash n}(\boldsymbol{\theta}) \tag{22}$$

(3) Building a new approximated posterior $q^\star$, estimating its parameters by matching moments with the tilted distribution:

$$\mathbb{E}_{q^*}(\phi(\boldsymbol{\theta})) = \mathbb{E}_{\hat{p}_n}(\phi(\boldsymbol{\theta})) \tag{23}$$

(4) Updating the approximated posterior $f_n$ by removing context factors from the new approximated posterior:

$$f_n(\boldsymbol{\theta}) \propto q^\star(\boldsymbol{\theta})/q^{\backslash n}(\boldsymbol{\theta}) \tag{24}$$

### A.2  CONDITIONAL MOMENT MATCH

With complex data likelihood where multiple variables interleave, such as the Tucker-based likelihood in the main paper, computation of $\mathbb{E}_{\hat{p}}(\phi(\boldsymbol{\Theta}))$ in moment match step of classic EP is intractable. To solve this problem, (Wang and Zhe, 2019) proposed a conditional moment matching method to update the parameters of the approximated posterior $f_n(\mathbf{Z}^k(i_k^n))$. Take $\mathbf{Z}^k(i_k^n)$ for an example, the required moments are $\phi(\mathbf{Z}^k(i_k^n)) = (\mathbf{Z}^k(i_k^n), \mathbf{Z}^k(i_k^n)\mathbf{Z}^k(i_k^n)^T)$. the key idea of CEP is to decompose the expectation into a nested structure:

$$\mathbb{E}_{\hat{p}}(\phi(\boldsymbol{\Theta})) = \mathbb{E}_{\hat{p}\left(\boldsymbol{\Theta}_{\backslash \mathbf{Z}^k(i_k^n)}\right)} \left[ \mathbb{E}_{\hat{p}\left(\mathbf{Z}^k(i_k^n)|\boldsymbol{\Theta}_{\backslash \mathbf{Z}^k(i_k^n)}\right)} \left[ \phi(\boldsymbol{\Theta}) \mid \boldsymbol{\Theta}_{\backslash \mathbf{Z}^k(i_k^n)} \right] \right], \tag{25}$$

where $\boldsymbol{\Theta}_{\backslash \mathbf{Z}^k(i_k^n)} \triangleq \boldsymbol{\Theta} \backslash \{\mathbf{Z}^k(i_k^n)\}$. Therefore, we can compute the conditional moment first, i.e., the inner expectation, with an analytical form. However, the computation of outer-expectation are under the marginal tilted distribution which is still intractable. To solve this problem, we follow the BCTT

(Fang et al., 2022) and apply a multivariate delta method  (Bickel and Doksum, 2015; Fang et al., 2021) to get:

$$\mathbb{E}_{q\left(\boldsymbol{\Theta}_{\backslash \mathbf{z}^k(i_k^n)}\right)}\left[\boldsymbol{\rho}_n\right] \approx \boldsymbol{\rho}_n\left(\mathbb{E}_q\left[\boldsymbol{\Theta}_{\backslash \mathbf{z}^k(i_k^n)}\right]\right) \tag{26}$$

where $\boldsymbol{\rho}_n$ denote the conditional moments of tilde distribution.

## A.3 CEP UPDATE FOR FUNBAT AND FUNBAT-CP

Following the above paradigm, we can work out the updating formulas for all parameters of the approximated message factors $\{f_n\} = \{\{f_n(\mathbf{Z}^k(i_k^n))\}_{k=1}^K, f_n(\tau)\}$ for FunBaT and FunBaT-CP.

For $f_n(\mathbf{Z}^k(i_k^n)) = \mathcal{N}(\mathbf{H}\mathbf{Z}^k(i_k^n)|\mathbf{m}_n^k, \mathbf{S}_n^k)$:

$$\mathbf{S}_n^k = (\mathbb{E}_q[\tau]\mathbb{E}_q[\mathbf{a}_n^{\backslash k}\mathbf{a}_n^{\backslash k^T}])^{-1}; \ \mathbf{m}_n^k = \mathbf{S}_n^k(y_n\mathbb{E}_q[\tau]\mathbb{E}_q[\mathbf{a}_n^{\backslash k}]), \tag{27}$$

where

$$\mathbf{a}_n^{\backslash k} = \mathcal{W}_{(k)}\left(\mathbf{U}^K(i_K^n) \otimes \ldots \otimes \mathbf{U}^{k+1}(i_{k+1}^n) \otimes \mathbf{U}^{k-1}(i_{k-1}^n) \otimes \ldots \otimes \mathbf{U}^1(i_1^n)\right), \tag{28}$$

and $\mathcal{W}_{(k)}$ is the folded tucker core $\mathcal{W}$ at mode $k$.

For $f_n(\tau) = \mathrm{Gam}(\tau|\alpha_n, \beta_n)$, the updating formulas are:

$$\alpha_n = \frac{3}{2}; \ \beta_n = \frac{1}{2}y_n^2 - y_n\mathbb{E}_q[\mathbf{a}_n] + \frac{1}{2}\mathrm{trace}[\mathbb{E}_q[\mathbf{a}_n\mathbf{a}_n^T]] \tag{29}$$

where:

$$\mathbf{a}_n = \mathrm{vec}(\mathcal{W})^T\left(\mathbf{U}^1(i_1^n) \otimes \ldots \otimes \mathbf{U}^K(i_K^n)\right) \tag{30}$$

The updating formulas of $f_n(\mathcal{W}) = \mathcal{N}(\mathrm{vec}(\mathcal{W}) \mid \mu_n, \mathcal{S}_n)$ is:

$$\mathbf{S}_n = (\mathbb{E}_q[\tau]\mathbb{E}_q[\mathbf{b}_n\mathbf{b}_n^T])^{-1}; \ \mu_n = \mathbf{S}_n(y_n\mathbb{E}_q[\tau]\mathbb{E}_q[\mathbf{b}_n]) \tag{31}$$

Where

$$\mathbf{b}_n = \mathbf{U}^1(i_1^n) \otimes \ldots \otimes \mathbf{U}^K(i_K^n). \tag{32}$$

We can apply the above update formulas for FunBaT-CP by setting the $\mathcal{W}$ as a constant diagonal tensor. However, We can also re-derive a more convenient and elegant form based on the Hadmard product form of CP. It will result in the similar formats on the update formulas of $f_n(\tau)$ and $f_n(\mathbf{Z}^k(i_k^n))$, but with the different definitions on $\mathbf{a}_n^{\backslash k}$ and $\mathbf{a}_n$. They are:

$$\mathbf{a}_n^{\backslash k} = \mathbf{U}^1(i_1^n) \circ \cdots \mathbf{U}^{k-1}(i_{k-1}^n) \circ \mathbf{U}^{k+1}(i_{k+1}^n) \cdots \circ \mathbf{U}^K(i_K^n) \tag{33}$$

and

$$\mathbf{a}_n = \mathbf{U}^1(i_1^n) \circ \cdots \circ \mathbf{U}^K(i_K^n) \tag{34}$$

## A.4 DERIVATION OF THE PROBABILISTIC IMPUTATION AT ANY INDEX

To derive the probabilistic imputation equation (17) in the main paper, we consider a general state space model, which includes a sequence of states $\mathbf{x}_1, \ldots, \mathbf{x}_M$ and the observed data $\mathcal{D}$. The states are at time $t_1, \ldots, t_M$ respectively. The key of the state space model is that the prior of the states is a Markov chain. The joint probability has the following form,

$$p(\mathbf{x}_1, \ldots, \mathbf{x}_M, \mathcal{D}) = p(\mathbf{x}_1) \prod_{j=1}^{M-1} p(\mathbf{x}_{j+1}|\mathbf{x}_j) \cdot p(\mathcal{D}|\mathbf{x}_1, \ldots, \mathbf{x}_M). \tag{35}$$

Note that here we do not assume the data likelihood is factorized over each state, like those typically used in Kalman filtering. In our point process model, the likelihood often couples multiple states together.

Suppose we have run some posterior inference to obtain the posterior of these states $q(\mathbf{x}_1, \ldots, \mathbf{x}_M)$, and we can easily pick up the marginal posterior of each state and each pair of the states. Now we want to calculate the posterior distribution of the state at time $t^*$ such that $t_m < t^* < t_{m+1}$. Denote the corresponding state by $\mathbf{x}^*$, our goal is to compute $p(\mathbf{x}^*|\mathcal{D})$. To do so, we consider incorporating $\mathbf{x}^*$ in the joint probability (35),

$$p(\mathbf{x}_1, \ldots, \mathbf{x}_m, \mathbf{x}^*, \mathbf{x}_{m+1}, \ldots, \mathbf{x}_M, \mathcal{D})$$
$$= p(\mathbf{x}_1) \prod_{j=1}^{m-1} p(\mathbf{x}_{j+1}|\mathbf{x}_j) \cdot p(\mathbf{x}^*|\mathbf{x}_m)p(\mathbf{x}_{m+1}|\mathbf{x}^*) \cdot \prod_{j=m+1}^{M} p(\mathbf{x}_{j+1}|\mathbf{x}_j) \cdot p(\mathcal{D}|\mathbf{x}_1, \ldots, \mathbf{x}_M). \quad (36)$$

Now, we marginalize out $\mathbf{x}_{1:M\setminus\{m,m+1\}} = \{\mathbf{x}_1, \ldots, \mathbf{x}_{m-1}, \mathbf{x}_{m+2}, \ldots, \mathbf{x}_M\}$. Note that since $\mathbf{x}^*$ does not appear in the likelihood, we can take it out from the integral,

$$p(\mathbf{x}_m, \mathbf{x}_{m+1}, \mathbf{x}^*, \mathcal{D})$$
$$= \int p(\mathbf{x}_1) \prod_{j=1}^{m-1} p(\mathbf{x}_{j+1}|\mathbf{x}_j) \prod_{j=m+1}^{M} p(\mathbf{x}_{j+1}|\mathbf{x}_j) \cdot p(\mathcal{D}|\mathbf{x}_1, \ldots, \mathbf{x}_M) \mathrm{d}\mathbf{x}_{1:M\setminus\{m,m+1\}}$$
$$\cdot p(\mathbf{x}^*|\mathbf{x}_m)p(\mathbf{x}_{m+1}|\mathbf{x}^*)$$
$$= \frac{p(\mathbf{x}_m, \mathbf{x}_{m+1}, \mathcal{D})p(\mathbf{x}^*|\mathbf{x}_m)p(\mathbf{x}_{m+1}|\mathbf{x}^*)}{p(\mathbf{x}_{m+1}|\mathbf{x}_m)}. \quad (37)$$

Therefore, we have

$$p(\mathbf{x}_m, \mathbf{x}_{m+1}, \mathbf{x}^*|\mathcal{D}) \propto p(\mathbf{x}_m, \mathbf{x}_{m+1}|\mathcal{D})p(\mathbf{x}^*|\mathbf{x}_m)p(\mathbf{x}_{m+1}|\mathbf{x}^*). \quad (38)$$

Suppose we are able to obtain $p(\mathbf{x}_m, \mathbf{x}_{m+1}|\mathcal{D}) \approx q(\mathbf{x}_m, \mathbf{x}_{m+1})$. We now need to obtain the posterior of $\mathbf{x}^*$. In the LTI SDE model, we know that the state transition is a Gaussian jump. Let us denote

$$p(\mathbf{x}^*|\mathbf{x}_m) = \mathcal{N}(\mathbf{x}^*|\mathbf{A}_1\mathbf{x}_m, \mathbf{Q}_1), \quad p(\mathbf{x}_{m+1}|\mathbf{x}_*) = \mathcal{N}(\mathbf{x}_{m+1}|\mathbf{A}_2\mathbf{x}^*, \mathbf{Q}_2).$$

We can simply merge the natural parameters of the two Gaussian and obtain

$$p(\mathbf{x}_m, \mathbf{x}_{m+1}, \mathbf{x}^*|\mathcal{D}) = p(\mathbf{x}_m, \mathbf{x}_{m+1}|\mathcal{D})\mathcal{N}(\mathbf{x}^*|\mathbf{m}^*, \mathbf{V}^*), \quad (39)$$

where

$$(\mathbf{V}^*)^{-1} = \mathbf{Q}_1^{-1} + \mathbf{A}_2^\top \mathbf{Q}_2^{-1} \mathbf{A}_2,$$
$$(\mathbf{V}^*)^{-1} \mathbf{m}^* = \mathbf{Q}_1^{-1} \mathbf{A}_1 \mathbf{x}_m + \mathbf{A}_2^\top \mathbf{Q}_2^{-1} \mathbf{x}_{m+1}. \quad (40)$$

## B   More discussion on the related work

**Functional Tensor Models**. Modeling the inner smoothness and continuity in tensor data in a functional manner has been a long-standing challenge and gets increasing attention in recent years. Early work like (Schmidt, 2009) uses GP with CP form to model the functional tensor, but lacks efficient inference to handle large-scale data. The community of low-rank approximation of black-box approximation has raised a series of work based on functional tensor-train(FTT), such as (Gorodetsky et al., 2015; Bigoni et al., 2016; Ballester-Ripoll et al., 2019; Chertkov et al., 2023). However, the series work of FTT depends on tensor-train format and polynomials-based approximation, which is not flexible enough and sensitive to hyperparameters. The similar idea of functional basis has also been used to model the smoothness in tensor decomposition (Imaizumi and Hayashi, 2017). Most recent work (Luo et al., 2023) also employs the Tucker format and uses the MLP to model the tensor mode functions and shows promising results. However, most of the existing models are purely deterministic and lack the probabilistic inference to handle data noise and uncertainty. In contrast, FunBaT is the first work to use the Tucker format to model the functional tensor in a probabilistic manner, and it enjoys the advantage of the linear-cost inference to handle large-scale data with uncertainty due to the usage of state-space GP.

**Differce between FunBaT, BCTT and SFTL**. From the technical perspective, BCTT (Fang et al., 2022) and SFTL (Fang et al., 2024) are the most similar work to FunBaT. Those methods utilize the state-space Gaussian Processes (GP) to model the latent dynamics in CP/Tucker decomposition

| Number of training samples | Observed Ratio | RMSE |
|---|---|---|
| 130 | 0.1 | 0.128 |
| 260 | 0.2 | 0.102 |
| 390 | 0.3 | 0.068 |
| 420 | 0.4 | 0.041 |
| 650 | 0.5 | 0.027 |
| 780 | 0.6 | 0.026 |
| 810 | 0.7 | 0.025 |

Table 3: Reconstruct loss of the synthetic data over different observed ratios.

and infer with message-passing techniques. This similarity has been noted in short in the related works section of the main paper, and we plan to highlight this more prominently in subsequent versions on their differences. The first difference is that BCTT and SFTL focus on time-series tensor data, whereas, FunBaT is centered on functional tensor data. This difference in application leads to distinct challenges and modeling: BCTT involves one Tucker-core dynamic with static factors, SFTL models time-varying trajectories factors and FunBAT employs a static core with groups of mode-wise dynamics. This fundamental difference in formulation leads to varied inferences. Due to the divergent formulations, the inference algorithms between the two methods show significant differences. For each observation, BCTT and SFTL need to infer only one state of the temporal dynamics, as they share the same timestamp, and the inference of all dynamics is synchronous. In contrast, FunBaT requires inferring multiple states of multiple dynamics, depending on each mode's index of the observation, which is more challenging. We will run a loop over tensor mode to do mode-wise conditional moment matching, and then get the message factors fed to different functions. The inference of multiple dynamics is asynchronous.

**Connection to Broader Coordinate-based Representation Model**. The idea of building parameterized models (MLP) to map the low-dimensional continuous coordinates to high-dimensional data voxel has boosted attention in recent years, especially in the scenarios of computer vision and graphics. The prior work CPNN (Tancik et al., 2020) tracks the challenges of classical "coordinate-based MLP" models and proposes the Fourier feature to improve the performance. Nerf (Mildenhall et al., 2021), one of the most crucial works in graphics recently, uses a large positional encoding MLP to reconstruct continuous 3D scenes from a series of 2D images, which utilizes the Fourier features of the spatial coordinates to better capture the high frequency. FunBaT could be seen as a generalization of these coordinate-based models to the tensor format, The main difference is that FunBaT has dimensional-wise functional representation, which means the mode-wise functions are independent and can be learned separately. This dimensional-wise representation fits the low-rank structure of the tensor data, and the learned mode-wise functions can be easily interpreted and visualized. We believe that the idea of FunBaT can be extended to the broader coordinate-based representation model and applications in CV and graphics, and we plan to explore this in future work.

## C    MORE EXPERIMENTS RESULTS

The reconstruction loss of the synthetic data over different observed ratios is shown in Table 3.

The prediction results on $R = \{3, 5, 7\}$ *BeijingAir-PM2.5*, *BeijingAir-PM10*, and *BeijingAir-SO2* are list in Tables 4 Tables 5,Tables 6

| | RMSE | | | MAE | | |
|---|---|---|---|---|---|---|
| Datasets | *PM2.5* | *PM10* | *SO2* | *PM2.5* | *PM10* | *SO2* |
| Discrete resolution:$50 \times 50 \times 150$ | | | | | | |
| P-Tucker | $0.812 \pm 0.054$ | $0.779 \pm 0.015$ | $0.668 \pm 0.015$ | $0.566 \pm 0.018$ | $0.56 \pm 0.013$ | $0.423 \pm 0.005$ |
| Tucker-ALS | $1.063 \pm 0.049$ | $1.036 \pm 0.062$ | $0.965 \pm 0.029$ | $0.727 \pm 0.021$ | $0.738 \pm 0.022$ | $0.628 \pm 0.01$ |
| Tucker-SVI | $0.758 \pm 0.017$ | $0.8 \pm 0.051$ | $0.652 \pm 0.014$ | $0.554 \pm 0.014$ | $0.583 \pm 0.017$ | $0.421 \pm 0.038$ |
| Discrete resolution:$100 \times 100 \times 300$ | | | | | | |
| P-Tucker | $0.794 \pm 0.099$ | $0.767 \pm 0.005$ | $0.683 \pm 0.043$ | $0.486 \pm 0.009$ | $0.512 \pm 0.014$ | $0.402 \pm 0.015$ |
| Tucker-ALS | $1.02 \pm 0.032$ | $1.011 \pm 0.023$ | $0.97 \pm 0.027$ | $0.738 \pm 0.011$ | $0.751 \pm 0.004$ | $0.681 \pm 0.016$ |
| Tucker-SVI | $0.681 \pm 0.027$ | $0.741 \pm 0.071$ | $0.714 \pm 0.128$ | $0.468 \pm 0.014$ | $0.526 \pm 0.043$ | $0.421 \pm 0.038$ |
| Discrete Resolution: $300 \times 300 \times 1000$ | | | | | | |
| P-Tucker | $1.493 \pm 0.125$ | $1.439 \pm 0.001$ | $1.264 \pm 0.2$ | $0.532 \pm 0.034$ | $0.575 \pm 0.001$ | $0.46 \pm 0.004$ |
| Tucker-ALS | $1.027 \pm 0.033$ | $1.027 \pm 0.038$ | $1.007 \pm 0.017$ | $0.743 \pm 0.01$ | $0.758 \pm 0.012$ | $0.699 \pm 0.007$ |
| Tucker-SVI | $1.657 \pm 0.135$ | $1.408 \pm 0.052$ | $1.451 \pm 0.062$ | $0.79 \pm 0.016$ | $0.768 \pm 0.016$ | $0.732 \pm 0.016$ |
| Discrete Resolution:$428 \times 501 \times 1461$ (original) | | | | | | |
| P-Tucker | $2.091 \pm 0.122$ | $2.316 \pm 0.001$ | $1.48 \pm 0.1$ | $0.756 \pm 0.002$ | $0.79 \pm 0.001$ | $0.556 \pm 0.017$ |
| Tucker-ALS | $1.008 \pm 0.013$ | $1.027 \pm 0.036$ | $1 \pm 0.023$ | $0.738 \pm 0.005$ | $0.757 \pm 0.009$ | $0.699 \pm 0.01$ |
| Tucker-SVI | $1.864 \pm 0.03$ | $1.686 \pm 0.061$ | $1.537 \pm 0.121$ | $0.885 \pm 0.016$ | $0.864 \pm 0.015$ | $0.787 \pm 0.032$ |
| methods using continuous indexes | | | | | | |
| FTT-ALS | $1.019 \pm 0.013$ | $1.001 \pm 0.013$ | $1.002 \pm 0.026$ | $0.744 \pm 0.007$ | $0.755 \pm 0.007$ | $0.696 \pm 0.011$ |
| FTT-ANOVA | $2.151 \pm 0.032$ | $2.006 \pm 0.015$ | $1.987 \pm 0.036$ | $1.788 \pm 0.031$ | $1.623 \pm 0.014$ | $1.499 \pm 0.018$ |
| FTT-cross | $0.943 \pm 0.026$ | $0.933 \pm 0.012$ | $0.845 \pm 0.026$ | $0.566 \pm 0.018$ | $0.561 \pm 0.011$ | $0.467 \pm 0.033$ |
| RBF-SVM | $0.995 \pm 0.015$ | $0.955 \pm 0.02$ | $0.794 \pm 0.026$ | $0.668 \pm 0.008$ | $0.674 \pm 0.014$ | $0.486 \pm 0.026$ |
| BLR | $0.998 \pm 0.013$ | $0.977 \pm 0.014$ | $0.837 \pm 0.021$ | $0.736 \pm 0.007$ | $0.739 \pm 0.008$ | $0.573 \pm 0.009$ |
| FunBaT-CP | $0.294 \pm 0.016$ | $\mathbf{0.347 \pm 0.036}$ | $\mathbf{0.384 \pm 0.01}$ | $\mathbf{0.183 \pm 0.006}$ | $0.236 \pm 0.014$ | $0.242 \pm 0.003$ |
| FunBaT | $\mathbf{0.291 \pm 0.017}$ | $0.348 \pm 0.036$ | $0.386 \pm 0.011$ | $\mathbf{0.183 \pm 0.01}$ | $\mathbf{0.233 \pm 0.012}$ | $\mathbf{0.241 \pm 0.004}$ |

Table 4: Prediction error over *BeijingAir-PM2.5*, *BeijingAir-PM10*, and *BeijingAir-SO2* with $R = 3$, which were averaged over five runs.

| | RMSE | | | MAE | | |
|---|---|---|---|---|---|---|
| Datasets | *PM2.5* | *PM10* | *SO2* | *PM2.5* | *PM10* | *SO2* |
| Discrete resolution:$50 \times 50 \times 150$ | | | | | | |
| P-Tucker | $0.835 \pm 0.078$ | $0.787 \pm 0.077$ | $0.745 \pm 0.046$ | $0.54 \pm 0.007$ | $0.535 \pm 0.007$ | $0.424 \pm 0.004$ |
| Tucker-ALS | $1.178 \pm 0.055$ | $1.123 \pm 0.033$ | $0.975 \pm 0.029$ | $0.741 \pm 0.014$ | $0.741 \pm 0.009$ | $0.615 \pm 0.006$ |
| Tucker-SVI | $0.738 \pm 0.022$ | $0.747 \pm 0.009$ | $0.638 \pm 0.01$ | $0.518 \pm 0.011$ | $0.541 \pm 0.007$ | $0.405 \pm 0.022$ |
| Discrete resolution:$100 \times 100 \times 300$ | | | | | | |
| P-Tucker | $0.808 \pm 0.065$ | $0.827 \pm 0.012$ | $0.763 \pm 0.023$ | $0.474 \pm 0.014$ | $0.494 \pm 0.008$ | $0.426 \pm 0.005$ |
| Tucker-ALS | $1.07 \pm 0.03$ | $1.038 \pm 0.022$ | $0.953 \pm 0.015$ | $0.745 \pm 0.01$ | $0.756 \pm 0.008$ | $0.675 \pm 0.007$ |
| Tucker-SVI | $0.768 \pm 0.105$ | $0.79 \pm 0.085$ | $0.691 \pm 0.087$ | $0.471 \pm 0.038$ | $0.524 \pm 0.035$ | $0.405 \pm 0.022$ |
| Discrete Resolution: $300 \times 300 \times 1000$ | | | | | | |
| P-Tucker | $2.153 \pm 0.271$ | $1.972 \pm 0.001$ | $1.486 \pm 0.054$ | $0.784 \pm 0.054$ | $0.859 \pm 0.001$ | $0.624 \pm 0.02$ |
| Tucker-ALS | $1.062 \pm 0.031$ | $1.046 \pm 0.029$ | $1.007 \pm 0.02$ | $0.747 \pm 0.01$ | $0.76 \pm 0.01$ | $0.699 \pm 0.007$ |
| Tucker-SVI | $1.584 \pm 0.092$ | $1.446 \pm 0.035$ | $1.511 \pm 0.065$ | $0.828 \pm 0.037$ | $0.805 \pm 0.012$ | $0.781 \pm 0.026$ |
| Discrete Resolution:$428 \times 501 \times 1461$ (original) | | | | | | |
| P-Tucker | $2.359 \pm 0.078$ | $2.426 \pm 0.001$ | $1.881 \pm 0.054$ | $1.011 \pm 0.021$ | $1.094 \pm 0.001$ | $0.775 \pm 0.017$ |
| Tucker-ALS | $1.045 \pm 0.02$ | $1.056 \pm 0.021$ | $0.999 \pm 0.025$ | $0.74 \pm 0.005$ | $0.761 \pm 0.007$ | $0.698 \pm 0.01$ |
| Tucker-SVI | $1.574 \pm 0.088$ | $1.603 \pm 0.024$ | $1.536 \pm 0.05$ | $0.842 \pm 0.026$ | $0.879 \pm 0.008$ | $0.812 \pm 0.018$ |
| methods using continuous indexes | | | | | | |
| FTT-ALS | $1.019 \pm 0.013$ | $1.000 \pm 0.013$ | $1.001 \pm 0.026$ | $0.744 \pm 0.007$ | $0.754 \pm 0.005$ | $0.695 \pm 0.010$ |
| FTT-ANOVA | $2.149 \pm 0.033$ | $2.006 \pm 0.014$ | $1.987 \pm 0.036$ | $1.788 \pm 0.031$ | $1.623 \pm 0.014$ | $1.499 \pm 0.018$ |
| FTT-cross | $0.941 \pm 0.024$ | $0.933 \pm 0.012$ | $0.831 \pm 0.015$ | $0.563 \pm 0.018$ | $0.560 \pm 0.011$ | $0.464 \pm 0.033$ |
| RBF-SVM | $0.995 \pm 0.015$ | $0.955 \pm 0.02$ | $0.794 \pm 0.026$ | $0.668 \pm 0.008$ | $0.674 \pm 0.014$ | $0.486 \pm 0.026$ |
| BLR | $0.998 \pm 0.013$ | $0.977 \pm 0.014$ | $0.837 \pm 0.021$ | $0.736 \pm 0.007$ | $0.739 \pm 0.008$ | $0.573 \pm 0.009$ |
| FunBaT-CP | $0.292 \pm 0.013$ | $0.352 \pm 0.035$ | $\mathbf{0.385 \pm 0.009}$ | $\mathbf{0.183 \pm 0.007}$ | $0.236 \pm 0.013$ | $0.243 \pm 0.003$ |
| FunBaT | $\mathbf{0.288 \pm 0.012}$ | $\mathbf{0.338 \pm 0.03}$ | $0.388 \pm 0.003$ | $0.191 \pm 0.021$ | $\mathbf{0.231 \pm 0.005}$ | $\mathbf{0.241 \pm 0.003}$ |

Table 5: Prediction error over *BeijingAir-PM2.5*, *BeijingAir-PM10*, and *BeijingAir-SO2* with $R = 5$, which were averaged over five runs.

| | RMSE | | | MAE | | |
|---|---|---|---|---|---|---|
| Datasets | *PM2.5* | *PM10* | *SO2* | *PM2.5* | *PM10* | *SO2* |
| Discrete resolution:$50 \times 50 \times 150$ | | | | | | |
| P-Tucker | $0.832 \pm 0.037$ | $0.823 \pm 0.112$ | $0.811 \pm 0.058$ | $0.521 \pm 0.005$ | $0.529 \pm 0.009$ | $0.425 \pm 0.005$ |
| GPTF | $0.694 \pm 0.013$ | $0.702 \pm 0.005$ | $0.607 \pm 0.016$ | $0.488 \pm 0.012$ | $0.501 \pm 0.005$ | $0.381 \pm 0.009$ |
| CP-ALS | $1.385 \pm 0.209$ | $1.092 \pm 0.04$ | $1.016 \pm 0.087$ | $0.804 \pm 0.048$ | $0.751 \pm 0.013$ | $0.635 \pm 0.031$ |
| Tucker-ALS | $1.379 \pm 0.046$ | $1.314 \pm 0.047$ | $1.049 \pm 0.025$ | $0.779 \pm 0.014$ | $0.775 \pm 0.015$ | $0.621 \pm 0.01$ |
| Tucker-SVI | $0.742 \pm 0.023$ | $0.721 \pm 0.01$ | $0.67 \pm 0.027$ | $0.504 \pm 0.014$ | $0.508 \pm 0.004$ | $0.408 \pm 0.011$ |
| Discrete resolution:$100 \times 100 \times 300$ | | | | | | |
| P-Tucker | $1.015 \pm 0.063$ | $0.911 \pm 0.063$ | $0.877 \pm 0.039$ | $0.51 \pm 0.012$ | $0.521 \pm 0.018$ | $0.452 \pm 0.009$ |
| Tucker-ALS | $1.086 \pm 0.041$ | $1.047 \pm 0.015$ | $0.967 \pm 0.029$ | $0.745 \pm 0.013$ | $0.755 \pm 0.007$ | $0.682 \pm 0.011$ |
| Tucker-SVI | $0.783 \pm 0.054$ | $0.787 \pm 0.052$ | $0.702 \pm 0.054$ | $0.464 \pm 0.017$ | $0.515 \pm 0.021$ | $0.408 \pm 0.011$ |
| Discrete Resolution: $300 \times 300 \times 1000$ | | | | | | |
| P-Tucker | $1.718 \pm 0.155$ | $1.928 \pm 0.001$ | $1.629 \pm 0.05$ | $0.805 \pm 0.043$ | $0.954 \pm 0.001$ | $0.687 \pm 0.006$ |
| Tucker-ALS | $1.073 \pm 0.039$ | $1.062 \pm 0.02$ | $1.007 \pm 0.018$ | $0.748 \pm 0.01$ | $0.762 \pm 0.01$ | $0.699 \pm 0.006$ |
| Tucker-SVI | $1.437 \pm 0.051$ | $1.499 \pm 0.027$ | $1.389 \pm 0.042$ | $0.793 \pm 0.032$ | $0.842 \pm 0.013$ | $0.772 \pm 0.018$ |
| Discrete Resolution:$428 \times 501 \times 1461$ (original) | | | | | | |
| P-Tucker | $2.134 \pm 0.174$ | $2.483 \pm 0.001$ | $2.001 \pm 0.149$ | $0.924 \pm 0.044$ | $1.055 \pm 0.001$ | $0.786 \pm 0.012$ |
| Tucker-ALS | $1.051 \pm 0.02$ | $1.054 \pm 0.029$ | $1 \pm 0.021$ | $0.741 \pm 0.005$ | $0.76 \pm 0.008$ | $0.698 \pm 0.01$ |
| Tucker-SVI | $1.292 \pm 0.021$ | $1.521 \pm 0.095$ | $1.454 \pm 0.054$ | $0.737 \pm 0.015$ | $0.872 \pm 0.027$ | $0.817 \pm 0.028$ |
| Methods using continuous indexes | | | | | | |
| FTT-ALS | $1.019 \pm 0.013$ | $1.000 \pm 0.013$ | $1.001 \pm 0.026$ | $0.744 \pm 0.007$ | $0.754 \pm 0.005$ | $0.695 \pm 0.010$ |
| FTT-ANOVA | $2.149 \pm 0.033$ | $2.006 \pm 0.014$ | $1.987 \pm 0.036$ | $1.788 \pm 0.031$ | $1.623 \pm 0.014$ | $1.499 \pm 0.018$ |
| FTT-cross | $0.941 \pm 0.024$ | $0.933 \pm 0.012$ | $0.831 \pm 0.015$ | $0.563 \pm 0.018$ | $0.560 \pm 0.011$ | $0.464 \pm 0.033$ |
| RBF-SVM | $0.995 \pm 0.015$ | $0.955 \pm 0.02$ | $0.794 \pm 0.026$ | $0.668 \pm 0.008$ | $0.674 \pm 0.014$ | $0.486 \pm 0.026$ |
| BL | $0.998 \pm 0.013$ | $0.977 \pm 0.014$ | $0.837 \pm 0.021$ | $0.736 \pm 0.007$ | $0.739 \pm 0.008$ | $0.573 \pm 0.009$ |
| PCMT-CP | $\mathbf{0.296 \pm 0.023}$ | $\mathbf{0.335 \pm 0.02}$ | $\mathbf{0.385 \pm 0.008}$ | $\mathbf{0.184 \pm 0.009}$ | $\mathbf{0.231 \pm 0.007}$ | $\mathbf{0.242 \pm 0.003}$ |
| PCMT-Tucker | $0.318 \pm 0.014$ | $0.347 \pm 0.017$ | $0.39 \pm 0.009$ | $0.198 \pm 0.008$ | $0.235 \pm 0.008$ | $\mathbf{0.242 \pm 0.003}$ |

Table 6: Prediction error over *BeijingAir-PM2.5*, *BeijingAir-PM10*, and *BeijingAir-SO2* with $R = 7$, which were averaged over five runs.

