# OpenReview forum: "Functional Bayesian Tucker Decomposition for Continuous-indexed Tensor Data"
_ICLR.cc/2024/Conference — ICLR 2024 poster_

### Official Review · Reviewer_TaJq · 2023-10-14

**Soundness:** 2 fair
**Presentation:** 2 fair
**Contribution:** 3 good
**Rating:** 5
**Confidence:** 4

**Summary:**

This paper introduces the Functional Bayesian Tucker Decomposition (FunBaT)
algorithm. Instead of treating the factors of the Tucker decomposition as
matrices with discrete row indexing, the authors instead think of the factors
as latent Gaussian process function priors. Earlier work in this area of
[Schmidt, ICML 2009] considers a simpler "continuous CP decomposition" for
tensors.
The authors then make connections
to stochastic differential equations (SDEs) and conditional expectation
propagation (CEP) for the learning task, which is somewhat analogous to a
continuous version of alternating least squares. The authors provide experiments
on synthetic and real-world datasets, of order $K=2$ and $K=3$ respectively.

**Strengths:**

- Extends work from Tucker to CP, since CP is a generalization of Tucker if you
  restrict the core tensor to be diagonal
- A "continuous" version of Tucker decomposition on the factors is an excellent
  idea to explore
- Uses interesting real-world datasets with continuous features (indices), and
  the results are comprehensive and compelling. That said, the US-TEMP
  experiment could be improved since a core shape of $(1,1,1)$ is used, which
  means a product of the factor functions is being learned.

**Weaknesses:**

- There is not enough emphasis on related work, especially with [Fang et al.,
  ICML 2022], which appears to be a *very similar paper*. This discussion should happen
  much earlier in the paper, and there should be more than one mention of this
  Fang et al. (2022), as it it almost gets buried in its current form. Missing
  seminal work:
  * "Bayesian Tensor Regression" by [Guhaniyogi et al., JMLR 2017]
- While the synthetic experiments have the potential to be very strong and
  compelling, it would be useful to see how the tensor reconstruction looks as a
  function of the number of samples. It may not be surprising in its current
  form that we can recover ground truth with $650$ samples.
- In the Beijing Air experiments, please explain if non-uniform core shapes were
  explored (as they might better help fit the data). A recent ICML paper
  investigates core shape selection and could be of interest:
  * "Approximately Optimal Core Shapes for Tensor Decompositions" by [Ghadiri
    et al., ICML 2023]

**Questions:**

### Questions
- [page 01] Is Tucker decomposition really a more compact low-rank
  representation? Maybe the factor matrices can be more compact, but doesn't
  this come at the cost of a larger "core structure" then, e.g., CP
  decomposition?
- [page 03] By "successive derivatives of $m$-th order", do you instead mean to
  write $\frac{d^m f(x)}{dx^{m}}$?
- [page 03] "the trained model cannot handle new objects with never-seen
  indices" -- this is not necessarily true, and is the core idea behind tensor
  completion.
- [page 04] Why do you use $\tau^{-1}$ as Gaussian noise (i.e., variance)
  instead of $\sigma^2$?
- [page 06] Is the preset (scalar) mode rank $R$ for a given Tucker core of
  shape $(r_1, \dots, r_K)$ the product? I.e., $R = \prod_{i=1}^K r_i$? If so,
  this should be stated explicitly and you need to remove the claim that "the
  linear costs of both time and space ..." since $R$ is exponential in the order
  $K$.

### Typos and suggestions
- [page 01] "there were finite" --> "there are finite"
- [page 01] "tensor train(TT)" --> "tensor train (TT)"
- [page 01] "Tucker decomposition is famous for ..." --> "Tucker decomposition
  is widely-used for ..."
- [page 01] "and time.." --> "and time."
- [page 01] "or simpler CP format" --> "or the simpler CP format"
- [page 01] "which is with more compact ..." --> "which is a more compact ..."
- [page 02] Inconsistent subheading capitalization (see ICLR style guide and
  example paper)
- [page 02] "under the following settings" --> "under the following setting"
- [page 02] typo at end of sentence with "groups of latent factors" (should be
  a comma instead of a period)
- [page 02] suggestion: consider superscripting factor matrices/vectors as
  ${U}^{(k)}$ instead of ${U}^k$ to remove any ambiguity in meaning
- [page 02] "The classic CANDECOMP/PARAFAC (CP)" --> "The classic CP" (you
  already introduced the acronym)
- [page 02] missing space: factors: $y_{i}$ (same for many other sentences on
  this page)
- [page 03] "denote as $f \sim$" --> "denoted as $f \sim$"
- [page 03] consider using $\ell$ instead of $l$ as a hyperparameter in the
  Matern kernel
- [page 03] missing space: "efficient $O(n)$ inference"
- [page 03] No need to give the meaning of "FunBaT" again
- [page 04] "preset latent rank:$\{r_1,\dots,r_K\}$ --> "preset latent rank
  $(r_1, \dots, r_K)$. Missing space between subsequent sentences.
- [page 04] suggestion: when you shift to "continuous-indexed tensors", it may
  be more clear to say that $(i_1^{n},\dots, i_K^{n}) \in \mathbb{R}^{K}$
- [page 04] missing space: "Namely, "
- [page 05] typo: "massage merging" --> "message merging"
- [page 06] punctuation typos in Algorithm 1 inputs
- [page 06] suggestion: Substantially more discussion / math can be given to
  FunBaT-CP. The focus of the paper is on Tucker decomposition, but this is a
  very nice special case for which it would be nice to know more of its
  properties.
- [page 06] Missing space: "function factorization is(Schmidt, 2009)." along
  with other citations in this paragraph.
- [page 07] typo: "alternating least square" --> "alternating least squares".
  Many more typos / punctuation errors in this paragraph.
- [page 09] typo: "lantide mode"

---

> ### Author Response · Authors · 2023-11-20
>
> We greatly thank and appreciate the reviewer's time and effort in offering insightful and detailed comments and reviews. We address the comments below:( C: comments; R: responses)
>
> >C1: "not enough emphasis on related similar work **BCTT** (Fang, et al. Bayesian Continuous-Time Tucker Decomposition, ICML 2022)", "Missing seminal work: "Bayesian Tensor Regression" by [Guhaniyogi et al., JMLR 2017]"
>
> R1: Good point. We acknowledge the use of similar techniques **BCTT**, specifically the utilization of state-space Gaussian Processes (GP) to model the latent dynamics in Tucker decomposition. As this similarity has been noted in short in the related works section of our paper, we do plan to follow your suggestion and highlight this more prominently in subsequent versions on their **significant differences in formulation and inference. Please refer to the response (R1) to reviewer MPRT for more detailed elaborations**. For the "Bayesian Tensor Regression"[Guhaniyogi et al., JMLR 2017], we will add it in the citations and related works section.
>
> >C2: "For the synthetic experiments,  it would be useful to see the tensor reconstruction looks as a function of the number of samples?"
>
> R2: Good suggestion! We add reconstruction RMSE with different numbers of samples, here are the results(we generate 1300 samples on the surface of the 3D function, and randomly select parts of samples with Gaussian noise (STD=0.02) for training, evaluate on the remaining samples:):
>
> |  Number of training samples   | Observation Ratio |   RMSE |
> | :--- |---: |---:  |
> | 130   | 0.1   | 0.128   |
> | 260  | 0.2      | 0.102  |
> | 390  |   0.3    | 0.068   |
> | 420 |   0.4    | 0.041   |
> | 650 |   0.5    | 0.027   |
> | 780 |   0.6    | 0.026   |
> | 810 |   0.7    | 0.025   |
>
> As we can see, the RMSE decreases with the increase of the number of training samples. With 650 samples, the RMSE is less than 0.03, which is similar to the noise level and can be seen as a good reconstruction. We will add the results in the latest version.
>
> >C3: explain if non-uniform core shapes were explored in the Beijing Air experiments with "Approximately Optimal Core Shapes for Tensor Decompositions" by [Ghadiri et al., ICML 2023]
>
> R3: Thanks for raising the great reference! We will add the discussion on non-uniform core in our experiments with the latest work [Ghadiri et al., ICML 2023].
>
> >C4: "Is Tucker decomposition really a more compact low-rank representation?"
>
> R4: Good point! We do agree that CP is more compact and that statement may be not accurate enough. We will polish it in the latest version. For a more detailed discussion on the compactness between Tucker  CP and TT, please refer to the response (R4) to reviewer MPRT.
>
> > C5: By "successive derivatives of
> m-th order", do you instead mean to write $d^m f(x)/ dx^m$?
>
> R5: Yes, we mean $d^m f(x)/ dx^m$. We will polish it to make it more clear.
>
> > C6: "The trained model cannot handle new objects with never-seen indices" -- this is not necessarily true, and is the core idea behind tensor completion.""
>
> R6: Our statement was intended to highlight that traditional tensor models typically rely on grid-structured data characterized by discrete indices, and thus may struggle with grid-free interpolation for new objects identified by real-valued indices. We apology for any confusion caused and will revise our wording to clarify this point more effectively.
>
> > C7: "why use $\tau^{-1}$ instead of $\sigma^2$ for the Gaussian noise?"
>
> R7: Good question. In the Bayesian framework we employ, the most common modeling approach for Gaussian noise involves assuming that its inverse: $\tau$ —also known as 'precision'— follows a Gamma distribution(see statement under equation 11). This convention is widely used in Bayesian analysis for Gaussian likelihood with observation noise."
>
> > C8: Is the preset (scalar) mode rank $R$
>  for a given Tucker core of shape $(r_1, \ldots, r_K)$
>  the product? I.e., $R = \prod_{k=1}^K r_k$?
>
> R8: No, actually the preset (scalar) mode rank $R$ is rank for each mode and the shape of core is $(R, R, \ldots, R)$. We will add the clarification in the latest version. For the detailed statement for linear cost with rank $R$, please refer to the **response (R4) for reviewer BN7i**
>
> >C9: Typos and suggestions
>
> R9: Thanks for the correction! We do greatly appreciate the reviewer's time and effort in offering such detailed help. We will fix them in the latest version.

---

> ### Comment · Reviewer_TaJq · 2023-11-22
>
> I have read the author response and am willing to increase my rating from 3 to 5.

---

> ### Author Response · Authors · 2023-11-22
>
> We thank you so much for reading our response and increasing the score!

---

### Official Review · Reviewer_BN7i · 2023-10-31

**Soundness:** 3 good
**Presentation:** 2 fair
**Contribution:** 2 fair
**Rating:** 6
**Confidence:** 3

**Summary:**

This paper proposed a function Tucker decomposition for tensors with continuous-valued indices. The authors firstly use a Gaussian process to map indices to Tucker factors and then contract these factors to obtain the entry value. To efficiently learn the GP prior, an algorithm based on state-space GP and expectation propagation is derived. For experiments, the authors test the model on synthetic data and several spatiotemporal data imputation tasks.

**Strengths:**

The authors study the functional tensor decomposition for continuously-indexed tensors. This seems to be an interesting and novel topic in the field tensor decomposition and may have some new applications.

**Weaknesses:**

1. The setting of continuously-indexed tensors is new in the community of tensor decomposition. However, my main concern is how this task is related and different from traditional regression tasks. Why do we need such construction of tensors and what is the significance of the proposed function Tucker decomposition. Considering the experiments, the authors show applications in spatiotemporal data imputation. However, there are many existing methods, including interpolation like GP, VAE [1], GAN [2], LSTM [3], diffusion models [4] and many of their variants. Besides, the problem setting is very similar to CPNN [5]. For a better understanding of the paper, I think it might be better to have a discussion of these lines of works and empirical comparisons
- [1]. Mattei, & Frellsen. (2019). MIWAE: Deep generative modelling and imputation of incomplete data sets. ICML.
- [2]. Yoon, et al. (2018). Gain: Missing data imputation using generative adversarial nets. ICML.
- [3]. Cao, et al. (2018). Brits: Bidirectional recurrent imputation for time series. NeurIPS
- [4]. Tashiro, et al. (2021). Csdi: Conditional score-based diffusion models for probabilistic time series imputation. NeurIPS.
- [5]. Tancik, et al. (2020). Fourier features let networks learn high frequency functions in low dimensional domains. NeurIPS.

2. Since the proposed model employs nonlinear GPs, it might be better to compare with some nonlinear or GP-based tensor decompositions. Also, baselines for continuous-time tensor decompositions are also good choices, as the authors also mentioned in the related work.

**Questions:**

1. Compared with GP regression, the main difference of the proposed model is preserving the Tucker structure in the final layer. I am wondering why this construction is so helpful as shown in empirical results.
2. There are matrix inversion and multiplications in the update rules. Why is the time complexity linear with the rank $R$?
3. The authors adopted three resolution settings for BeijingAir datasets. What is the setting for US-TEMP?

typo: In paragraph above Figure 1, whore tensor -> whole tensor?

---

> ### Author Response · Authors · 2023-11-20
>
> We thank the reviewer for the careful review! We address the comments below:( C: comments; R: responses)
>
> >C1: "Relation and Differences Between Functional Tensor Models and Traditional Regression Tasks and Models [1,2,3,4,5].
>
> R1: We are grateful to the reviewer for posing such an insightful question and for the excellent references provided, particularly CPNN [5]. We will certainly cite these and expand our discussion on this topic in the latest version of our manuscript.
>
> A brief and high-level response is that, in addition to **learning a good mapping from indexes to values**—the focus of traditional regression tasks and models—**functional tensor models** emphasize **learning robust representations of latent dynamics**. This approach not only facilitates more flexible and robust prediction but also enhances interpretability and utility for downstream tasks.
>
> Consider the US-TEMP dataset as an illustrative example. Each observation entry represents a temperature value corresponding to specific spatial-temporal indexes, such as (time, latitude, longitude). This is a quintessential spatial-temporal dataset. Various model types, ranging from simple linear regression to advanced deep models like CNNs, GNNs, VAEs, and diffusion models, can handle this task. However, these models are typically **black-box** in nature, offering limited insight into latent dynamics and lacking flexibility for potential downstream tasks. For instance, predicting temperature at a new location with additional features, like *altitude*, would typically require retraining the model from scratch.
>
> Conversely, our proposed method is capable of learning latent dynamics and providing a **low-rank, dynamic representation** of key factors within the data. As demonstrated in Figure 2, the learned latent dynamics not only yield accurate predictions but also help elucidate the evolving processes underlying the data. We further emphasize the potential of these representations for downstream tasks. Taking the example of incorporating *altitude* as a new feature, our method allows for the addition of a new mode to the tensor. We can reuse the learned representations of existing features and only train the new mode dynamics. These **reusable representations** are crucial for achieving more flexible and general modeling capabilities."
>
> > C2: Compare with some nonlinear or GP-based tensor decompositions and continuous-time tensor decompositions.
>
> R2: Good suggestion! We add the comparison with GP-based tensor decomposition: **GPTF**(Zhe, et al. Distributed flexible nonlinear
> tensor factorization, 2016)  and continuous-time tensor decomposition: **BCTT**(Fang, et al.Bayesian Continuous-Time Tucker Decomposition, 2022) on the three datasets of BeijingAir. The mean of RMSE over 5 runs are shown:
> <!-- add tables -->
>
> |  PM2.5   | R=2 |   R=3 |     R=5 |    R=7 |
> | :---        |            ---: |---:  | ---:  | ---:  |
> | **GPTF**   | 0.424   | 0.425   |  0.477  | 0.464   |
> |**BCTT**  | 0.375      | 0.372  | 0.386   | 0.371    |
> | **FunBaT-CP**  |   0.296    | 0.294   | 0.292   | **0.296**   |
> | **FunBaT** |   **0.288**    | **0.291**   | **0.288**   | 0.318   |
>
> |  PM10    | R=2 |   R=3 |     R=5 |    R=7 |
> | :---        |            ---: |---:  | ---:  | ---:  |
> | **GPTF**   | 0.454  | 0.479   |  0.535  | 0.531   |
> |**BCTT**  | 0.408      | 0.421  | 0.443   | 0.420    |
> | **FunBaT-CP**  |   0.343    | **0.347**   | 0.352   | **0.335**   |
> | **FunBaT** |   **0.328**    | 0.348   | **0.338**  | 0.347   |
>
> |  SO2   | R=2 |   R=3 |     R=5 |    R=7 |
> | :---        |            ---: |---:  | ---:  | ---:  |
> | **GPTF**   | 0.459   | 0.475   |  0.540  | 0.504   |
> |**BCTT**  | 0.411      | 0.418  | 0.398   | 0.410    |
> | **FunBaT-CP**  |   **0.386**    | **0.384**   | **0.385**   | **0.385**   |
> | **FunBaT** |   **0.386**    | 0.386   | 0.388   | 0.390   |
>
>  As we can see, for every rank, **FunBaT** is with the best performance. The **BCTT** gets the second-best performance, as it can model the continuous dynamics for time mode, but not for other modes. The **GPTF** is with the worst performance. The reason is that **FunBaT** is a with-nature model for functional tensor data, as we discussed in C1-R1.

---

> ### Author Response · Authors · 2023-11-20
>
> > C3: Why the construction of Tucker is helpful compared with GP regression?
>
> Good question! We claim that our model actually **wraps multiple GP regressions with a tensor structure(Tucker or CP)**. In the case of a $K$-mode tensor with rank $R$, our approach involves managing $K \times R$ separate GP regressions. This is in stark contrast to a conventional GP regression, which typically relies on a single kernel to capture the entirety of multi-view data patterns. The tensor-based design **splits the high-order data and multi-view interactions into low-rank, distinct subspaces**. This decomposition **significantly simplifies the learning process for each individual GP**. The concept of decomposing and representing complex, high-dimensional data in a low-rank format is a broadly adopted strategy in the field of machine learning."
>
> >C4: Why is the time complexity linear with the rank $R$?
>
> R4: Great point! The state-space GP is assigned to each dimension of the mode independently (equation 7). Thus, for mode with rank $R$, we will have $R$ separate state-space GP, and each GP's state dimension—determining the matrix mat/inverse cost in KF/RTS— is only related to kernel-decided order $m$ ($m=1$ for $Matern-3/2$  kernel, $m=2$ for $Matern-5/2$  kernel).  During the inference over specific mode,  **we run the KF and RTS over the $R$ separate state-space dynamics, saying, a loop over $R$ to run KF and RTS.** Thus, the time complexity is linear with the rank $R$.
>
> We do understand the confusion is that we use the compact notation $Z^k$ to refer the concatenated states of all $R$ dimensions. We will add the clarification in the latest version and thanks again for pointing out this point.
>
> > C5:  Why not put a multi-resolution setting for US-TEMP?
>
> R5: Good point! We do not add the multi-resolution setting for US-TEMP is because the data's original size is relatively small: (15 latitudes, 95 longitudes, 267 timestamps), compared with BeijingAir, whose size is (428 atmospheric pressure, 501 temperature, 1461 timestamps). Discretization will make the data even smaller, and not suitable for real-world applications. We will add the discussion on this in the latest version.
>
> > C6:  Typo in the paragraph above Figure 1
>
> R6: Thanks for the correction! We will fix it in the latest version.

---

> > ### Comment · Reviewer_BN7i · 2023-11-23
> >
> > Thanks for the authors to address my concerns. I would like to raise my score to 6.

---

> > > ### Author Response · Authors · 2023-11-23
> > >
> > > We thank you so much for reading our response and increasing the score!

---

### Official Review · Reviewer_wp4E · 2023-11-01

**Soundness:** 3 good
**Presentation:** 3 good
**Contribution:** 3 good
**Rating:** 8
**Confidence:** 4

**Summary:**

The paper proposed an Bayesian method for tensor Tucker decomposition, where the data are assumed to have continues index. Each column of the factor matrices is modeled using Gaussian process (GP), and efficient method was proposed to infer the parameters of the GP. The idea of this paper is quite interesting,
and the experimental results are convincing.

**Strengths:**

1, the idea that model factor matrices using GP is new.
2, efficient inference methods are proposed to determine the unknown parameters.

**Weaknesses:**

1, the methods need to set the rank of the tensor manually, which is usually unknown in practice.
3, In table 1, the MSE of the proposed method much lower than the baselines. But no explanation is provided to show why.

**Questions:**

1, Does the equation 4 implies the smoothness of z(x)?
2, Compared to Tucker decomposition, CP decomposition offers several advantageous properties, particularly its uniqueness, making it more favored in many applications. However, it is worth noting that CP decomposition typically requires larger factor matrices compared to Tucker decomposition. Therefore, it is interesting to discuss the computational complexity of the proposed method in relation to the rank of the tensor and evaluate its applicability in scenarios that require setting a large rank.

---

> ### Author Response · Authors · 2023-11-20
>
> We thank reviewer wp4E for the valuable comments and support! We address the comments below:( C: comments; R: responses)
>
> > C1: Explanation of MSE of the proposed method much lower than the baselines in Table 1.
>
>  R1: Good question! For the significant performence gain of the proposed method in Table 1, we think the main reasons are:
> 1.  The three pollution tensor datasets have inherent and strong **mode-wise continuity**, as their modes are : (pressure, temperature, time). The proposed method can capture the mode-wise continuity by the **mode-wise dynamics**. However, the baselines, like all static tensor methods, can not model that fact.
> 2.  The observations of the three datasets are **sparse**. The observation ratio is less than 0.001%. The FTT series baseline, with **too many basis functions**, is easy to get **overfitting** issue, as it is with  The proposed method is with **robust** modeling, as it is with probabilistic inference. The Bayesian framework can be seen as a **regularization** to avoid overfitting.
>
> > C2: Equation 4 implies the smoothness of z(x)?
>
>  R2: "Good point! As equation 4 indicates,  z(x)  is a Linear Time-Invariant (LTI) Stochastic Differential Equation (SDE), so we cannot say it's **smooth** under the strict mathematical definition. In fact, we cannot label any SDE or stochastic process as **smooth**, as stochastic processes can be unbounded. However, we can state that z(x)  is **stationary**. Given that z(x)  is an LTI SDE, its mean and autocovariance function are time-invariant. Generally speaking, with its stationary characteristics, the curve of z(x)  is unlikely to exhibit significant abrupt changes, meaning that its curve tends to be relatively smooth.
>
>
> > C3: Discussion on the computational complexity over rank for CP and Tucker.
>
>  R3: Good point! We do agree the trade-off between the efficiency of CP and the capacity of Tucker is tricky and crucial for real-world applications. We will add a discussion on this in the future version.

---

### Official Review · Reviewer_MPRT · 2023-11-01

**Soundness:** 3 good
**Presentation:** 3 good
**Contribution:** 2 fair
**Rating:** 5
**Confidence:** 4

**Summary:**

The paper proposes a method called Functional Bayesian Tucker Decomposition (FunBaT) to handle continuous-indexed tensor data. Traditional tensor decomposition methods are designed for discrete and finite-dimensional indexes, but real-world data often contains continuous indexes such as geographic coordinates. FunBaT solves this problem by treating the continuous-indexed data as the interaction between the Tucker core and a group of latent functions. Gaussian processes are used as functional priors to model the latent functions, and an equivalent stochastic differential equation is used to reduce computational cost. The paper introduces an efficient inference algorithm based on advanced message-passing techniques. FunBaT outperforms existing methods in synthetic and real-world applications while being able to identify interpretable patterns. The paper also provides explanations on tensor decomposition, function factorization, the use of Gaussian processes as state-space models, and the FunBaT model and algorithm. Overall, the paper offers a solution for tensor decomposition on continuous-indexed tensor data.

**Strengths:**

1. Extension to continuous-indexed tensor data: The paper proposes a method called Functional Bayesian Tucker Decomposition (FunBaT) that extends traditional Tucker decomposition to handle continuous-indexed tensor data.
2. Utilization of Gaussian processes as functional priors: FunBaT models the latent functions in continuous-indexed data using Gaussian processes as functional priors. This allows for flexible and efficient modeling of the latent functions. Additionally, the paper reduces computational cost by constructing an equivalent stochastic differential equation, further enhancing the efficiency of the method.
3. Improved performance and interpretable patterns: FunBaT is demonstrated to outperform existing methods in both synthetic and real-world applications. It not only achieves lower prediction errors but also has the ability to identify interpretable patterns in the data.

**Weaknesses:**

1. The paper's innovation and significance may not be convincingly conveyed. The idea of using Gaussian processes to model N factor matrices for continuous data appears to be straightforward and intuitive. Furthermore, the paper's modeling approach is very similar to that of reference [3], which employs Gaussian processes to model core tensors.
2. There are now many function decomposition models based on Tucker decomposition, e.g., [1,2], so the Introduction section regarding related works appears to be inaccurate.
3. Furthermore, the significance of this paper isn't fully articulated. Merely stating that there is currently no function Tucker decomposition doesn't seem to provide a sufficiently compelling reason for its importance.
4. Regarding the optimization algorithm, this paper is also closely related to some previous work, e.g., [3]
[1] M.Imaizumi, K.Hayashi (2017). "Tensor Decomposition with Smoothness". PMLR: International Conference on Machine Learning 2017
[2] Luo Y, Zhao X, Li Z, et al. Low-Rank Tensor Function Representation for Multi-Dimensional Data Recovery[J]. arXiv preprint arXiv:2212.00262, 2022.
[3] Fang S, Narayan A, Kirby R, et al. Bayesian Continuous-Time Tucker Decomposition[C]//International Conference on Machine Learning. PMLR, 2022: 6235-6245.

**Questions:**

1. What are the advantages of this function tensor decomposition method compared to existing approaches? Utilizing the Tucker decomposition allows for the representation of higher-order tensors and effectively mitigates the problem of the curse of dimensionality. Why does the Tucker decomposition lead to a more compact and flexible low-rank representation compared with TT and CP?
2. What is the primary motivation behind this paper in comparison to other function CP, Tucker, and TT decompositions? If this method was established solely because GP-based functional Tucker decomposition does not currently exist, I believe the motivation appears to be rather weak.
3. What is the primary innovation of this paper? Apart from considering continuous tensor indices in the modeling, is there a deeper level of differences compared to Bayesian Continuous Tucker Decomposition?

---

> ### Author Response · Authors · 2023-11-20
>
> We thank reviewer MPRT for their careful review! We address the comments below:( C: comments; R: responses)
>
> >C1: “**FunBAT** modeling approach is very similar to that of **BCTT** (Bayesian Continuous-Time Tucker Decomposition, ICML 2022)” , What is the primary innovation of **FunBAT**? What's the deeper level of differences between **FunBAT** and **BCTT**.
>
> R1: Good point! We acknowledge the use of similar techniques **BCTT**, specifically the utilization of state-space Gaussian Processes (GP) to model the latent dynamics in Tucker decomposition. This similarity has been noted in short  in the related works section of our paper, and we plan to highlight this more prominently in subsequent versions on their differences:
>
> - **Differences in target scenarios and formulations**:   **BCTT** focuses on time-series tensor data, whereas ,**FunBAT** is centered on functional tensor data. This difference in application leads to distinct challenges and modeling:  **BCTT** involves **one Tucker-core dynamic** with static factors, **FunBAT** employs a static core with **groups of mode-wise dynamics**. This fundamental difference in formulation leads to varied inferences.
>
> - **Differences in Inference Algorithms**:
>  Due to the divergent formulations, the inference algorithms between the two methods show significant differences. **For each observation**, **BCTT** needs to infer only **one state of the single temporal dynamic**. In contrast, **FunBaT** requires inferring **multiple states of multiple dynamics**, depending on each mode's index of the observation. which is more challenging. This results in distinctly different procedures in terms of message approximation, passing, and merging. Specifically, in **BCTT**, all Tucker core time-varying functions share the same timestamp, and then can be treated as **one temporal dynamic** by stacking. Thus, we only need to approximate the message once, and then the dynamic inference is in a **synchronous** manner. However, in **FunBaT**,  we will run a loop over tensor mode to do mode-wise conditional moment matching, and then get the messages factors fed to different functions. The inferece of multiple dynamics is in a **asynchronous** manner.
>
> >C2: Existing functional Tucker decomposition works e.g., [1,2]
>
> R2: Thanks for the excellent reference. We will modify our claim. We will add them in citations and correct the statements.
>
> >C3: What are the advantages of this function tensor decomposition method compared to existing approaches? What's the primary motivation of this paper? Is it only because "there is no functional Trucker?"
>
> R3: Great question! The main advantage of **FunBAT** is that **it's a scalable and flexible Bayesian model**, with inherent abilities for robust modeling and probabilistic inference. For example, it can **offer uncertainty-aware prediction at any index (equation 17)**. However, most existing works on functional tensor taking **deterministic approximation**, saying, the weighted sum of basis functions(FTT series, [1]), or MLP ([2]), can not do this.
>
> The Bayesian methods, featured for robustness and uncertainty quantification, are underused in this field. The only work in the early year [Schmidt,2009] that bridged the Bayesian method and function tensor, is not scalable and limited to CP. Thus, as there is  **no scable Bayesian method for functional tensor**, we design **FunBAT** by utilizing advanced Bayesian learning tools, like CEP[Wang,2019] and state-space GP[Solin,2016]. That is our primary innovation and motivation.
>
>  We also want to claim **the proposed Bayesian framework is not only limited for Tucker, it also supports CP(section 4.2)**. We highlight the "Functional Tucker" as Tucker formulation is more challenging.  Furthermore, we believe the proposed framework can be extended to TT too, which will form a general scalable Bayesian framework fitting various decomposition methods. We will explore more in-depth in this direction in our future work.
>
> >C4:  Why does the Tucker decomposition lead to a more compact and flexible low-rank representation compared with TT and CP?
>
> R4: Good point! In our view, compared with CP, Tucker is with stronger capacity as it models all possible interactions. Compared to TT, Tucker's rank setting is more flexible, as there is no constraint for the mode rank. However, TT sets a chain of matrix multiplication to get tensor entry, constraining the size(rank) of adjacent mode to fit the valid matrix multiplication. Moreover, the compact core tensor of Tucker can provide insights of the interpretability of which interactions are crucial, it will help people better understand the data.  On the other side, we acknowledge that the statement may be not accurate enough and ambiguous, and will polish it in the latest version.

---

> > ### Author Response · Authors · 2023-11-23
> >
> > Dear Reviewer MPRT,
> >
> > We greatly appreciate the time you took to review our paper. Due to the short duration of the author-reviewer discussion phase, we would appreciate your feedback on whether your main concerns have been adequately addressed. We are ready and willing to provide further explanations and clarifications if necessary.
> >
> > Thank you very much!

---

### Author Response · Authors · 2023-11-21
**Invitation for a Discussion**

We appreciate all reviewers' time and efforts in evaluating our work! In view of the limited available time, we would kindly like to ask the reviewers to please engage in a discussion with us (if not) given the submitted rebuttals so we can respond to the possible new questions in time.

---

### Meta-Review · Area_Chair_GDq8 · 2023-12-06

**Metareview:**

This paper generalizes Tucker decomposition to data that does not fal neatly into a grid structure. This is important in applications where the indices are continuous such as geographically indexed data. The four reviewers highlighted several strengths including the main contribution - an extension to continuous indexed data, and the predictive performance in experiments. Some reviewers were concerned that the existing literature was not sufficiently discussed in order to put this work into context. The authors offered changed that I think will improve the presentation of the paper. The paper presents well-reasoned experiments to address two questions: "What is the benefit of using continuous indices compared to a finer discrete grid?", "Does this use of continuous indices reveal more useful information about the data tha discrete grid?", and "Is the computational cost of using a continuous index bearable for the benefits?" The authors clearly addressed these pertinent questions in the experiments even given limited space.

**Justification For Why Not Higher Score:**

The paper reviews are not uniformly above the acceptance threshold.

**Justification For Why Not Lower Score:**

The reviewers judged the paper to be innovative and the experiments to be well-reasoned. The large-scale data application is relevant to many scientific domains and the method seems to improve on a discrete Tucker decomposition framework which is not obvious.

---

### Decision · Program_Chairs · 2024-01-16

Accept (poster)